MS-TP-22-57

# *Gravitational waves from the early universe*

Rafael R. Lino dos Santos[1,2,æ*] and Linda M. van Manen [3,å]

**1** CP3-Origins, University of Southern Denmark, Campusvej 55, DK-5230 Odense M, Denmark
**2** University of Münster, Institute for Theoretical Physics, 48149 Münster, Germany
**3** Friedrich-Schiller-Universität, Institute for Theoretical Physics, 07743 Jena, Germany
æ rado@cp3.sdu.dk å linda.van.manen@uni-jena.de

October 30, 2023

## Abstract

**Ongoing and future gravitational wave collaborations explore different frequency ranges of the gravitational wave spectrum, probing different stages of the early universe and Beyond Standard Model physics. Due to the very high energies involved, accelerators cannot probe these earlier stages. Therefore, until some years ago, knowledge about new physics was limited and relied on bounds from CMB observations and theoretical assumptions about higher energy scales. While models could be constrained by CMB data, they were left unconstrained at shorter wavelength scales. Nonetheless, each one of these models has a gravitational wave density spectrum at these shorter wavelength scales that can ultimately be compared to data from ground-based, space-born, and pulsar timing array searches. These lecture notes review the formalism of gravitational waves in General Relativity and introduce stochastic gravitational waves, with a focus on primordial sources and commenting on detection efforts.**

**These lecture notes were inspired by the course "Gravitational Waves from the Early Universe" given at the 27th W.E. Heraeus "Saalburg" Summer School 2021 by Valerie Domcke.**

# 1 Motivation

Look far away into the deep abyss of space and see how the first galaxy formed a long time ago. Look further and see the first stars. We can look all the way back to a time when the universe had a temperature of approximately $T \approx eV$, and free electrons for the first time combined with protons to form hydrogen. An event known as recombination.[1] After recombination, photons could travel freely through space. Since then, they have propagated in the universe,

---

[1]The term recombination mainly confuses students about the number of times the event of protons and electrons combining has occurred. This event has, as a matter of fact, occurred only once.

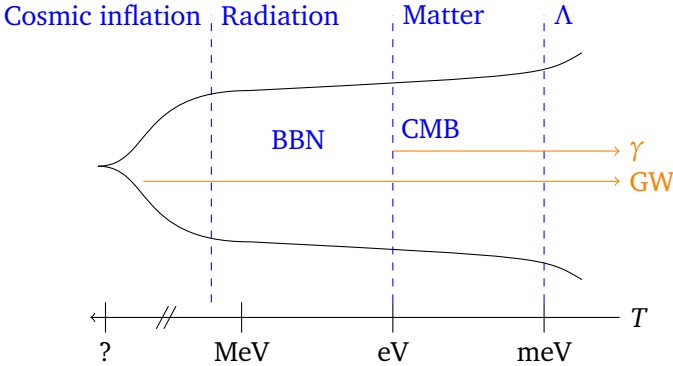

Figure 1: *A visualization of our cosmic history. Starting from the big bang on the left, our universe went through an era of cosmic inflation, followed by a radiation-dominated era, a matter-dominated era, and, finally, the current era dominated by dark-energy (cosmological constant Λ), which accelerates the expansion of the universe. The different eras are presented on a timescale given in the horizontal axis by T, the temperature of the universe that decreases over t, the time since the Big Bang. The characteristic energy scales of these different eras are also presented on the horizontal axis. The earliest freely moving photons we can observe today were imprinted in the cosmic microwave background (CMB) radiation after photon decoupling. The emission of these photons is represented by γ. On the other hand, gravitational waves (GW) from the early universe could be observed from much earlier times. They could be produced as early as cosmic inflation, are expected to travel across the universe largely unperturbed, and are expected to be detected in the coming decades, giving important hints for the development of inflation and new physics models.*

occasionally reaching our detectors. The moment when the first photons could freely travel through the universe is known as photon decoupling. These first photons are still visible today as the Cosmic Microwave Background (CMB), as a background noise from all directions, easily confused with pigeon poop on the antenna.[2] Before photon decoupling, light could not travel freely through the hot proton-electron plasma making up the young universe. The photons scattered continuously off the electrons and protons in the hot plasma, making the universe opaque. This shrouds everything that happened before photon decoupling in darkness and makes it complicated for physicists to observe what happened at the beginning and in the first years of our universe.

However, not all hope is lost with the discovery of gravitational waves. Early universe phenomena could have created gravitational waves. They may have been produced as early as cosmic inflation, creating a background of gravitational waves similar to the CMB, known as the gravitational wave background or stochastic gravitational wave background (SGWB). Unlike photons before decoupling, gravitational waves traveled through the early universe largely unperturbed. Thus, no fundamental obstacle prevents us from observing these early gravitational waves and discovering information about the earlier stages of our universe. Although there are no fundamental obstacles, there are plenty of experimental challenges.

Similar to the CMB, the gravitational wave background is expected to be like noise from all directions. The gravitational background noise is very weak and must be distinguished from other noise sources. Nonetheless, different collaborations expect to detect the gravitational

---

[2]When Robert W. Wilson and Arno A. Penzias first heard the radio signal, they initially thought it might have been caused by the poop of the pigeons surrounding the radio antenna. Not until they cleaned the antenna of the pigeon poop they realized what they had discovered.

wave background in the coming years,[3] and a lot can be learned from the early universe with gravitational waves. Take, for example, the gravitational wave frequency spectrum. Gravitational waves can come in different frequencies, and each collaboration probes a different spectrum range. Analogous to an orchestra where different instruments are combined, it is possible to combine data from different collaborations in a *gravitational wave orchestra* to get information about different stages of the universe.[4]

Finally, different early universe sources can produce gravitational waves. Examples are inflation and beyond Standard Model (BSM) physics phenomena, such as cosmic strings and first-order phase transitions. Remarkably, BSM depends on energy scales far beyond what accelerators on Earth can probe. Therefore, the gravitational wave background also serves as a laboratory to probe new physics! All this information in the gravitational wave background further completes our knowledge of our cosmic history.

The lectures are organized in the following sequence. In Sec. 2, we obtain gravitational waves as vacuum solutions of the linearized Einstein equations and study the effects of gravitational waves on test masses. Next, in Sec. 3, we study sourced emission of gravitational waves and their energy-momentum tensor and derive Einstein's quadrupole formula for the power emitted by a source. Then, in Sec. 4, we focus on the background of stochastic gravitational waves, derive the main properties of interferometers and PTA searches, and describe detection efforts. In Sec. 5, we give a very short review on cosmic inflation, explore properties of gravitational waves in the expanding Friedmann-Robertson-Lemaître-Walker universe, and show how data can be used to constrain BSM physics. Finally, in Sec. 6, we discuss some cosmological sources of gravitational waves (cosmic gravitational microwave background, single-field slow-roll inflation, axion-inflation, scalar-induced gravitational waves, first-order phase transitions, and cosmic strings). We conclude in Sec. 7.

## 2 Linearized Einstein equations

In this first lecture, we start by evaluating linearized general relativity, which describes the dynamics of a slightly perturbed gravitational field. After all, we can think about gravitational waves as *small* ripples in flat spacetime. Hence, we consider a metric tensor decomposed into the Minkowski metric and a small perturbation,

$$g_{\mu\nu} = \eta_{\mu\nu} + h_{\mu\nu}(x), \quad \text{with } |h_{\mu\nu}| << 1, \tag{2.1}$$

where higher order in $h$ can be omitted due to the smallness of $h$. Furthermore, we use the $(-,+,+,+)$ sign notation for $\eta_{\mu\nu}$ and the indices are raised with $\eta_{\mu\nu}$, i.e., $g^{\mu\nu} = \eta^{\mu\nu} - h^{\mu\nu}$. Afterward, we will look into the number of degrees of freedom the metric perturbation contains and discuss the most used gauge for fixing the unphysical degrees of freedom. Lastly, we will solve the Einstein equation for test masses far from the source of gravitational waves. All the material in the first two lectures is based on the book "Gravitational Waves: Volume 1: Theory and Experiments" by Michele Maggiore [1] and "Spacetime and Geometry. An introduction to general relativity" by Sean Carroll [2]. We recommend these references for an elaborate and detailed explanation of linearized general relativity and gravitational waves.

---

[3]Some of these collaborations are already active. They rely on ground-based detectors (LIGO, Virgo, KAGRA) or pulsar time array collaborations (NANOGrav, EPTA, PPTA, CPTA, InPTA, IPTA). Future collaborations include a space-based detector (LISA) and ground-based detectors (Cosmic Explorer and the Einstein telescope).

[4]Following the analogy with the music world, in a string quintet we can go from the double bass (low) to the cello, and then to the viola and the violins (high). Likewise, in the gravitational wave orchestra of the early universe, we can go from matter domination to radiation domination era, then reheating and inflation.

## 2.1 The linearized Einstein equations

The familiar Einsteins equations are given by,

$$G_{\mu\nu} \equiv R_{\mu\nu} - \frac{1}{2}g_{\mu\nu}R = \frac{8\pi G}{c^4}T_{\mu\nu}, \tag{2.2}$$

which relates the spacetime geometry, encoded in the metric $g_{\mu\nu}$, to matter described by the energy-momentum tensor $T_{\mu\nu}$. The Ricci tensor $R_{\mu\nu}$ and Ricci scalar $R$ for the linearized theory are computed following the usual scheme, starting from the Christoffel symbol. One can easily check that the linearized Christoffel symbol is given by

$$\begin{aligned}
\Gamma^{\rho}_{\mu\nu} &= \frac{1}{2}g^{\rho\sigma}[\partial_{\mu}g_{\nu\sigma} + \partial_{\nu}g_{\mu\sigma} - \partial_{\sigma}g_{\mu\nu}] \\
&= \frac{1}{2}\eta^{\rho\sigma}[\partial_{\mu}h_{\nu\sigma} + \partial_{\nu}h_{\mu\sigma} - \partial_{\sigma}h_{\mu\nu}] + \mathcal{O}(h^2),
\end{aligned} \tag{2.3}$$

which leads to the following Riemann curvature tensor,

$$\begin{aligned}
R^{\mu}_{\nu\sigma\rho} &= \partial_{\sigma}\Gamma^{\mu}_{\nu\rho} - \partial_{\rho}\Gamma^{\mu}_{\nu\sigma} + \Gamma^{\mu}_{\sigma\lambda}\Gamma^{\lambda}_{\nu\rho} - \Gamma^{\mu}_{\rho\lambda}\Gamma^{\lambda}_{\nu\sigma} \\
&= \frac{1}{2}[\eta^{\mu\lambda}(\partial_{\sigma}\partial_{\nu}h_{\rho\lambda} - \partial_{\sigma}\partial_{\rho}h_{\nu\lambda} - \partial_{\sigma}\partial_{\lambda}h_{\nu\rho} - (\sigma \leftrightarrow \rho)] + \mathcal{O}(h^2).
\end{aligned} \tag{2.4}$$

Note that the $\Gamma^2$ terms are higher-order terms in $h$ and will not contribute to the first-order Einstein equations. With a bit of algebra, one can then find the Ricci tensor,

$$R_{\mu\nu} = R^{\rho}_{\mu\rho\nu} = \frac{1}{2}(\partial_{\rho}\partial_{\mu}h^{\rho}_{\nu} + \partial_{\rho}\partial_{\nu}h^{\rho}_{\mu} - \partial_{\mu}\partial_{\nu}h - \Box h_{\mu\nu}) + \mathcal{O}(h^2), \tag{2.5}$$

and the Ricci scalar

$$R = g^{\mu\nu}R_{\mu\nu} = \partial_{\mu}\partial_{\nu}h^{\mu\nu} - \Box h + \mathcal{O}(h^2), \tag{2.6}$$

with $h = h^{\mu}_{\mu}$ the trace and $\Box = \partial_{\mu}\partial^{\mu}$. Combining all the results gives us the linearized Einstein tensor

$$G_{\mu\nu} = \frac{-1}{2}[\Box h_{\mu\nu} + \eta_{\mu\nu}\partial^{\rho}\partial^{\sigma}h_{\rho\sigma} - \eta_{\mu\nu}\Box h - \partial^{\rho}\partial_{\nu}h_{\mu\rho} - \partial_{\rho}\partial_{\mu}h^{\rho}_{\nu} + \partial_{\nu}\partial_{\mu}h] + \mathcal{O}(h^2). \tag{2.7}$$

This is a rather lengthy equation; hence it is usually preferred to define the trace reversed quantity $\bar{h}_{\mu\nu} \equiv h_{\mu\nu} - \frac{1}{2}\eta_{\mu\nu}h$. This simplifies the equation a little to

$$G_{\mu\nu} = \frac{-1}{2}[\Box\bar{h}_{\mu\nu} + \eta_{\mu\nu}\partial^{\rho}\partial^{\sigma}\bar{h}_{\rho\sigma} - \partial^{\rho}\partial_{\nu}\bar{h}_{\mu\rho} - \partial^{\rho}\partial_{\mu}\bar{h}_{\nu\rho}] + \mathcal{O}(h^2). \tag{2.8}$$

## 2.2 Gauge conditions

General relativity is invariant under all coordinate transformations $x^{\mu} \to x'^{\mu}(x)$, where $x'^{\mu}(x)$ is an arbitrary function of $x^{\mu}$. The metric will transform under this coordinate transformation as

$$g_{\mu\nu} \to g_{\mu\nu}(x') = \frac{\partial x^{\alpha}}{\partial x'^{\mu}}\frac{\partial x^{\beta}}{\partial x'^{\nu}}g_{\alpha\beta}(x), \tag{2.9}$$

which is known as the *gauge symmetry* of GR, also known as diffeomorphisms.

   The linearized theory, however, is only invariant under infinitesimal coordinate transformations and finite, global Poincaré transformations. Here, we fix the Minkowski metric $\eta_{\mu\nu}$ and choose small coordinate transformations under which $h_{\mu\nu}$ slightly changes, although leaves

$\eta_{\mu\nu}$ unchanged. It is convenient to choose a fixed inertial coordinate system on the Minkowski background because the Minkowski background has a lot of rotational symmetries [2]. Hence, this allows us to decompose the perturbation $h_{\mu\nu}$ based on its transformation under *spatial* rotations on a hypersurface. Under these spatial rotations, the metric perturbation can be decomposed into scalars, vectors, and tensors, which transform independently from each other. This allows us to write the Einstein equations for the *linearized theory* as a set of uncoupled ordinary differential equations.

### 2.2.1 Scalar, vector, tensor (SVT) decomposition

Before discussing the gauge transformation, it is educational to have a closer look at the scalar, vector, tensor (SVT) decomposition. The metric perturbation $h_{\mu\nu}$ is a $(0, 2)$ tensor with a spatial $SO(3)$ symmetry. Under these rotations, the $h_{00}$ component is a scalar, $h_{0i}$ is a three-vector, and $h_{ij}$ is a spatial rank 2 symmetric tensor [2]. This tensor can further be decomposed into a trace and a trace-free part. In group theory language, these are the irreducible representations of the spatial rotation group.

Therefore, the components of the metric $g_{\mu\nu}$ can then be written as

$$
\begin{aligned}
g_{00} &= -(1 + 2\Phi), \\
g_{i0} &= g_{0i} = 2a(\partial_i B - S_i), \\
g_{ij} &= a^2[(1 - 2\Psi)\delta_{ij} + 2\partial_{ij}F + (\partial_i T_j + \partial_j T_i) + t_{ij}].
\end{aligned}
\tag{2.10}
$$

Here, $g_{00} = -1$, and $g_{ij} = a^2\delta_{ij}$ are components of the background metric. The remaining terms are part of the perturbation $h_{\mu\nu}$ consisting of 4 scalars $(\Phi, B, \Psi, F)$, 2 vectors $(S_i, T_i)$, and 1 tensor $(t_{ij})$, with

$$
\partial_i T^i = 0, \quad \partial_i S^i = 0, \quad t_i^i = 0, \quad \text{and} \quad \partial_i t_j^i = 0.
\tag{2.11}
$$

We find 10 independent functions in the decomposed metric. Namely, 4 of the scalars, 4 vector components, and 2 tensor components of the $3 \times 3$ symmetric tensor $t_{ij}$.

## 2.3 Gauge transformation and gauge fixing

Before we can start solving the linearized Einstein equations, we need to address the ambiguous definition of the perturbation $h_{\mu\nu}$. As described above, the metric perturbation may have different forms depending on the choice of coordinate system. Indeed, if we consider an infinitesimal coordinate transformation

$$
x'^\mu = x^\mu + \xi^\mu,
$$

then (2.9) tells us that the perturbation transforms as

$$
h_{\mu\nu} \rightarrow h'_{\mu\nu} = h_{\mu\nu} - \partial_\mu \xi_\nu - \partial_\mu \xi_\nu,
\tag{2.12}
$$

where $\xi$ is small such that the conditions for a linearized theory, $|h_{\mu\nu}| << 1$, is preserved. Since $h'_{\mu\nu}$ is also a solution to the linearized Einstein equation, this is known as the *gauge transformation* of the linearized theory and is entirely analogous to the gauge transformation in electrodynamics. That is, we know from electrodynamics that if the vector potentials $A_\mu$ solves the Maxwell equation, then $A'_\mu = A_\mu - \partial_\mu \psi$, with $\psi$ any scalar field, is also a solution. Hence, multiple potentials give rise to the same field strength, and thus the same physical effects. Similarly, the Riemann tensor, a physical quantity, can be constructed from different

metric tensors.

In general, the vector $\vec{\xi}$ can be written in terms of two scalars and one vector, $\xi^\mu = (\xi^0, \partial_i f + f_i)$, with $\partial_i f^i = 0$ [1]. Consequently, two scalars and one vector can be gauged away by fixing $\xi$. This reduces the independent components of $h_{\mu\nu}$ with two independent scalars, and the two degrees of freedom that were left in the vectors, are gauged away. This can easily be seen by checking that the scalar, vector, and tensor parts transform as

$$
\begin{aligned}
\Phi &\to \Phi + \partial_0 \xi_0 \\
B &\to B - \xi_0 - \partial_0 f \\
\Psi &\to \Psi + \frac{1}{3}\nabla^2 f \\
F &\to F - 2f \\
S_i &\to S_i - \partial_0 f_i \\
T_i &\to T_i - f_i \\
t_{ij} &\to t_{ij},
\end{aligned}
\tag{2.13}
$$

which also shows a gauge invariant tensor.

A convenient choice of gauge, which is commonly used to fix $\vec{\xi}$, is the Lorentz gauge, $\partial_\mu \bar{h}^{\mu\nu} = 0$ [1]. Note that we are using the trace reversed metric here. The Lorentz gauge is always applicable, as we will show. Assume an arbitrary perturbation for which $\partial^\mu \bar{h}_{\mu\nu} \neq 0$. Then under an infinitesimal coordinate transformation, this transforms as

$$
\partial^{\prime\mu} \bar{h}'_{\mu\nu}(x') = \partial^\mu \bar{h}_{\mu\nu}(x) - \Box \xi_\nu.
\tag{2.14}
$$

By simply choosing $\Box \xi_\nu = \partial^\mu \bar{h}_{\mu\nu}$ the term on the left side will become zero, i.e., $\partial^{\prime\mu} \bar{h}'_{\mu\nu}(x') = 0$. A solution can always be found since the d'Alembertian operator is invertible. Hence, by choosing the appropriate $\vec{\xi}$, any metric perturbation that initially does not obey the Lorentz gauge $\partial^\mu \bar{h}_{\mu\nu} \neq 0$, can always be written in the Lorentz gauge. Now that $\partial^{\prime\mu} \bar{h}'_{\mu\nu}(x') = 0$, we can always continue making gauge transformations,

$$
\begin{aligned}
\partial^{\prime\prime\mu} \bar{h}''_{\mu\nu}(x'') &= \partial^{\prime\mu} \bar{h}'_{\mu\nu}(x') - \Box \xi_\nu \tag{2.15} \\
&= \partial^\mu \bar{h}_{\mu\nu}(x) - 2\Box \xi_\nu \tag{2.16} \\
&= -\partial^\mu \bar{h}_{\mu\nu}(x). \tag{2.17}
\end{aligned}
$$

As one can see, to remain in the Lorentz gauge after this gauge transformation, the vector $\xi_\mu$ needs to satisfy $\Box \xi_\mu = 0$. Hence, from the 10 independent components which we started with, the Lorentz gauge $\partial^{\prime\mu} \bar{h}'_{\mu\nu}(x')$ removes 4 components, however, it leaves residual freedom for gauge transformations with $\Box \xi_\mu = 0$. Here, $\Box \xi$ depends on 4 independent arbitrary functions $\xi_\mu$, as described above, and will therefore remove 4 more components.

We choose $\xi_0$ such that the trace of the trace reversed metric is zero, i.e., $\bar{h}^\mu_\mu = 0$, and the functions $\xi_i$ are chosen such that $\bar{h}_{0i} = 0$. Note that by making the metric traceless we get $\bar{h}_{\mu\nu} = h_{\mu\nu}$. Furthermore, it follows that

$$
\partial^\mu \bar{h}_{\mu\nu} = \partial^0 \bar{h}_{00} + \partial^i \bar{h}_{0i} = 0,
$$

where we fixed $\bar{h}_{0i} = 0$. Thus we have $\partial^0 \bar{h}_{00} = 0$, i.e. $\bar{h}_{00}$ is a constant in time. This static part of the metric is the Newtonian potential of the source. The gravitational wave is given by

the time-dependent part of the metric and, as long as we are interested in gravitational waves, $\partial^0 \bar{h}_{00} = 0$ means $\bar{h}_{00} = 0$.

To summarize, we have set

$$h_{0\mu}^{TT} = 0, \quad h_i^{i\,TT} = 0, \quad \partial^j h_{ij}^{TT} = 0. \tag{2.18}$$

This is known as the transverse traceless gauge or TT gauge. After applying the TT gauge, we have reduced the symmetric metric with 10 degrees of freedom to only 2 degrees of freedom. These are the 2 degrees of freedom that describe the two polarizations of a gravitational wave. Furthermore, the linearized Einstein equations in terms of the trace-reversed metric have now been reduced to a simple expression:

$$\Box h_{\mu\nu}^{TT} = \frac{-16\pi G}{c^4} \Lambda_{\mu\nu,\rho\sigma} T_{\rho\sigma}, \tag{2.19}$$

where the lambda tensor $\Lambda_{\mu\nu,\rho\sigma}$ is the TT projector which shall be discussed further in the second lecture.

## 2.4  Vacuum solutions

Let us think about a gravitational wave detector far away from any gravitational wave source. Hence, in a vacuum where $T_{\mu\nu} = 0$, such that the Einstein equation reduces to

$$\Box h_{\mu\nu}^{TT} = 0. \tag{2.20}$$

This has a plane wave solution,

$$h_{\mu\nu}^{TT}(x) = A_{\mu\nu}(k) \sin k^\alpha x_\alpha, \tag{2.21}$$

where $k^\alpha = (\omega/c, \vec{k})$ is the wave vector, and $A_{\mu\nu}(k)$ is the polarization tensor, which contains information about the amplitude of the gravitational wave and the polarization properties. Due to the restrictions imposed by the TT gauge, we can conclude that the amplitude is traceless and purely spatial. What is left to ensure that we are in the *tranverse* traceless gauge, is to check if the perturbation is transverse. In other words,

$$\partial^\mu h_{\mu\nu}^{TT} = k^\mu A_{\mu\nu}(k) \sin k^\alpha x_\alpha = 0. \tag{2.22}$$

This relation is true if the wavevector is orthogonal to the polarization tensor, $k^\mu A_{\mu\nu} = 0$. For example [1], if the wave is propagating in the $\hat{z}$- direction, then $A_{z\nu} = 0$. Considering that $A_{0\nu} = A_i^i = 0$ and $A_{\mu\nu}$ is also symmetric, we can generally write

$$\begin{aligned} A_{xx} &= -A_{yy} \equiv h_+, \\ A_{xy} &= A_{yx} \equiv h_\times, \end{aligned} \tag{2.23}$$

and the rest is zero. Thus

$$A_{\mu\nu}(k) = \begin{pmatrix} 0 & 0 & 0 & 0 \\ 0 & h_+ & h_\times & 0 \\ 0 & h_\times & -h_+ & 0 \\ 0 & 0 & 0 & 0 \end{pmatrix}. \tag{2.24}$$

To check if this really is a solution we can plug it into the equation of motion:

$$\Box \bar{h}_{\mu\nu}^{TT} = k^\alpha k_\alpha A_{\mu\nu}(k) \sin k^\alpha x_\alpha = 0. \tag{2.25}$$

Note that not all components of $A_{\mu\nu}$ are zero, which means that

$$k^\alpha k_\alpha = 0 \;\rightarrow\; \omega^2 = c^2 |\vec{k}|^2. \tag{2.26}$$

This is a rough proof that gravitational waves travel at the speed of light!

## 2.5 Effects of gravitational waves on test masses

To examine the effect of gravitational waves on mass, let us first consider a single particle, with a geodesic trajectory parametrized by $x^\mu(\tau)$. The geodesic equation is given by

$$\frac{d^2 x^\mu}{d\tau^2} + \Gamma^\mu_{\rho\nu}(x) \frac{dx^\nu}{d\tau} \frac{dx^\rho}{d\tau} = 0. \tag{2.27}$$

We assume that the particle is approximately static, such that $\frac{dx^\mu}{d\tau} \approx (1,0,0,0)$, and we can assume that $\tau \approx x^0$. With these assumptions, the geodesic equation will be

$$\frac{d^2 x^\mu}{d\tau^2} = -(\dot{h}_{\mu 0} - \frac{1}{2} \partial_\mu h_{00}), \tag{2.28}$$

where the dot stands for the time derivative. After applying the TT gauge, we can see that in this gauge and at *linear order*, single particles are not affected by gravitational waves.

Now consider a second particle with a geodesic parametrized by $x^\mu(\tau) + \xi^\mu(\tau)$. This particle will satisfy the geodesic equation

$$\frac{d^2(x^\mu + \xi^\mu)}{d\tau^2} + \Gamma^\mu_{\rho\nu}(x + \xi) \frac{d(x^\nu + \xi^\nu)}{d\tau} \frac{d(x^\rho + \xi^\rho)}{d\tau} = 0. \tag{2.29}$$

It is assumed that $\xi$ is much smaller than the length scale of the gravitational waves, such that we can expand in $\xi$ to the first order. Then by taking the difference between the two geodesic equations we obtain the geodesic deviation equation,

$$\frac{d^2 \xi^\mu}{d\tau^2} + 2\Gamma^\mu_{\rho\nu}(x) \frac{dx^\nu}{d\tau} \frac{\xi^\rho}{d\tau} + \xi^\sigma \partial_\sigma \Gamma^\mu_{\nu\rho}(x) \frac{dx^\nu}{d\tau} \frac{x^\rho}{d\tau} = 0, \tag{2.30}$$

describing the motion of the test particles relative to each other. We choose coordinates such that the Christoffel symbol vanishes at the spacetime position of the first test point particle. The derivative of the Christoffel symbol, however, will not vanish.

Note that the geodesic deviation equation is written in our chosen coordinate system. The covariant derivative of $\xi^\mu$ is given by

$$\frac{D^2 \vec{\xi}}{D\tau^2} = \frac{d}{d\tau} \frac{d}{d\tau} (\xi^\mu \hat{e}_\mu) = \frac{d}{d\tau} \left( \frac{d\xi^\mu}{d\tau} \hat{e}_\mu + \xi^\mu \frac{d\hat{e}_\mu}{d\tau} \right) \tag{2.31}$$

$$= \frac{d}{d\tau} \left( \frac{d\xi^\nu}{d\tau} + \xi^\mu \Gamma^\nu_{\mu\rho}(x) \frac{dx^\rho}{d\tau} \right) \hat{e}_\nu \tag{2.32}$$

$$= \left( \frac{d^2 \xi^\nu}{d\tau^2} + \xi^\mu \partial_\sigma \Gamma^\nu_{\mu\rho}(x) \frac{dx^\sigma}{d\tau} \frac{dx^\rho}{d\tau} \right) \hat{e}_\nu \tag{2.33}$$

where we used $\Gamma^\nu_{\mu\rho} = \frac{d\hat{e}_\mu}{dx^\rho} \hat{e}^\nu$, and we choose our coordinates such that $\Gamma^\nu_{\mu\rho}(x) = 0$. By combining this equation with the equation (2.30), we obtain

$$\frac{d^2 \xi^\mu}{d\tau^2} + R^\mu_{\nu\sigma\rho} \frac{dx^\rho}{d\tau} \frac{dx^\nu}{d\tau} \xi^\sigma. \tag{2.34}$$

If we assume a non-relativistic motion of the test particles, thus $\frac{dx^i}{d\tau} << \frac{dx^0}{d\tau}$, and notice from equation (2.4) that $R^i_{0j0} = \frac{-1}{2c^2} \ddot{h}^i_j{}^{TT}$, the geodesic deviation equation can be reduced to

$$\ddot{\xi}^i = -c^2 R^i_{0j0} \xi^j = \frac{1}{2} \ddot{h}^i_j{}^{TT} \xi^j. \tag{2.35}$$

As an example [1], let us consider the + polarisation and study the motion of test particles in the $xy$ plane. In this case,

$$h_{ab}^{TT} = h_+ \sin \omega t \begin{pmatrix} 1 & 0 \\ 0 & -1 \end{pmatrix}, \quad a, b = \{x, y\}. \tag{2.36}$$

The distance between the particles can generally be written as

$$\xi_a(t) = (X_0 + \delta X(t), Y_0 + \delta Y(t)), \tag{2.37}$$

where $(X_0, Y_0)$ are the unperturbed coordinates and $\delta X(t), \delta Y(t)$ are the displacements from the gravitational waves. Equation (2.35) results in

$$\delta \ddot{X} = \frac{-h_+}{2}(X_0 + \delta X)\omega^2 \sin \omega t, \tag{2.38}$$

$$\delta \ddot{Y} = \frac{h_+}{2}(Y_0 + \delta Y)\omega^2 \sin \omega t, \tag{2.39}$$

where the linear terms $\delta X$ and $\delta Y$ on the right-hand side can be neglected, since $\delta X << X_0$ and $\delta Y << Y_0$. Integrating the equations and we get

$$\delta X = \frac{h_+}{2}X_0 \sin \omega t, \quad \delta Y = \frac{-h_+}{2}Y_0 \sin \omega t. \tag{2.40}$$

The result for a ring of test masses is shown in Fig. (2).

# 3 Emission of gravitational waves

Having solved the linearized Einstein equation for the vacuum case in the last lecture, in the second lecture we focus on the Einstein equation with a source term, i.e.,

$$\Box \bar{h}_{\mu\nu} = -\frac{16\pi G}{c^4} T_{\mu\nu}. \tag{3.1}$$

After having solved the equation above, we will discuss gravitational waves in a curved background and derive the energy-momentum tensor for gravitational waves. We close this lecture by deriving Einstein's quadrupole formula.

## 3.1 Gravitational waves emitted by source

Equation (3.1) is solved by using the Green function for the d'Alembertian operator $\Box$,

$$\Box_x G(x^\sigma - y^\sigma) = \delta^{(4)}(x^\sigma - y^\sigma), \tag{3.2}$$

where $x^\sigma$ and $y^\sigma$ are depicted in Fig. 3. This is exactly how it is done in the analogous electromagnetic problem. The general solution is

$$\bar{h}_{\mu\nu}(x^\sigma) = -\frac{16\pi G}{c^4} \int G(x^\sigma - y^\sigma) \, T_{\mu\nu}(y^0, \vec{y}) \, d^4 y, \tag{3.3}$$

with $G(x^\sigma - y^\sigma) = -\frac{1}{4\pi|\vec{x} - \vec{y}|} \delta[|\vec{x} - \vec{y}| - (x^0 - y^0)]\theta(x^0 - y^0)$, and the theta function equals one when $x^0 > y^0$ [2]. After integrating over $y^0$ we obtain

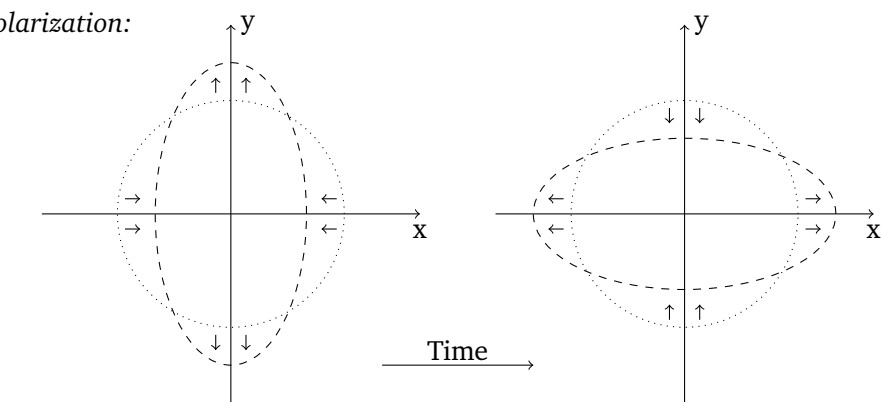

*+ polarization:*

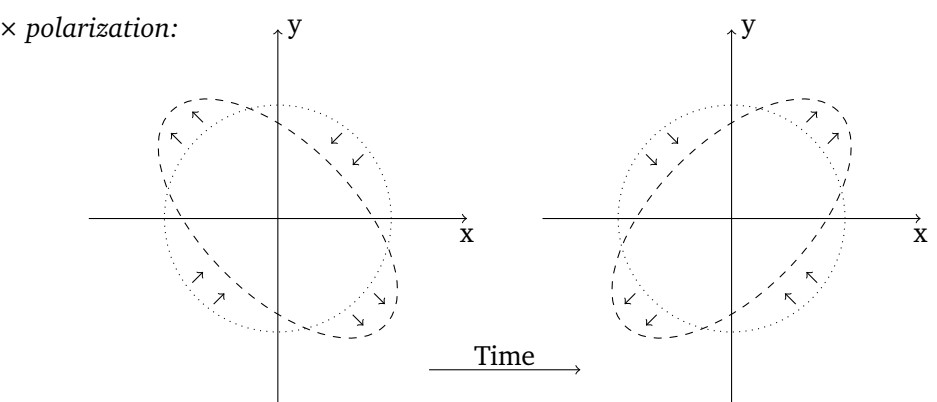

*× polarization:*

Figure 2: *A gravitational wave traveling in z-direction with a + polarization will curve spacetime such that a ring of test masses (gray dots) is alternating between a vertical and horizon elliptical shape, creating a + sign. A × polarized gravitational wave creates × sign as shown in the bottom figure.*

$$\bar{h}_{\mu\nu}(t,\vec{x}) = \frac{4G}{c^4} \int \frac{1}{|\vec{x}-\vec{y}|} T_{\mu\nu}(t-|\vec{x}-\vec{y}|)d^3y, \tag{3.4}$$

where $t = x^0$ and $t - |\vec{x} - \vec{y}| = t_r$ is referred to as the retarded time.

In the following, we will make the assumption that the source is far away and slowly moving. Hence the source is centered at a distance $\vec{x}$, and the edge of the source is at a distance $\vec{r} = \vec{x} - \vec{y}$, as is shown in Fig. 3. In terms of $r$, the gravitational wave takes the form

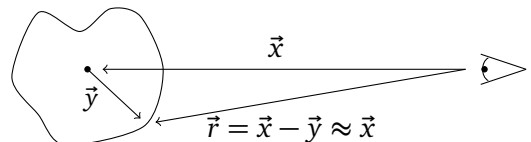

Figure 3: *An observer observes the center of a gravitational wave source at a distance $\vec{x}$. The outer edge of the source is at a distance $\vec{y}$ from the center and thus observed at a distance $\vec{r} = \vec{x} - \vec{y}$. The source is a large distance from the observer, hence $\vec{r} \approx \vec{x}$.*

$$\bar{h}_{\mu\nu}(t,\vec{x}) = \frac{4G}{rc^4} \int d^3y \; T_{\mu\nu}(t - \frac{r}{c}, y). \tag{3.5}$$

As we have seen before, in the vacuum case, the temporal components are set to zero, and thus we are only interested in the spatial components. This is given by

$$\int d^3y \; T_{ij} = \frac{1}{2} \partial_0^2 \int d^3y \; y_i y_j T_{00}(y). \tag{3.6}$$

To prove this relation, note that the energy/momentum conservation implies $\partial_\mu T^{\mu\nu} = 0$, and thus we can derive

$$\begin{aligned}
\partial_\mu T^{0\mu} &= \partial_0 T^{00} + \partial_k T^{0k} = 0 \\
\partial_0^2 T^{00} &= -\partial_k \partial_0 T^{0k} = \partial_k \partial_l T^{lk} \\
y_i y_j \partial_0^2 T^{00} &= y_i y_j \partial_k \partial_l T^{lk} = 2T_{ij}
\end{aligned} \tag{3.7}$$

In the second line, energy/momentum conservation was used again and the last step is obtained by partial integration and $\partial_k y_i = \delta_{ki}$. Thus

$$\bar{h}_{ij}(t,\vec{x}) = \frac{2G}{c^4} \frac{1}{r} \partial_0^2 \int d^3y \; y_i y_j T_{00}(t - \frac{r}{c}, y), \tag{3.8}$$

where it is conventional to define the integral as the tensor moment of the source $I_{ij}(t - \frac{r}{c})$. The resulting formula

$$\bar{h}_{ij}(t,\vec{x}) = \frac{2G}{c^4} \frac{1}{r} \partial_0^2 I_{ij}(t - \frac{r}{c}) \tag{3.9}$$

is known as the quadrupole formula.

The transverse traceless gauge for gravitational waves outside the sources and propagating in $\hat{n}$ direction is found by projecting the solution onto the TT gauge [1]. For this, we introduce a transverse projector $P_{ij} = \delta_{ij} - n_i n_j$, with $\hat{n} = \frac{\vec{k}}{k}$, which is used to construct the Lambda tensor $\Lambda_{ij,kl} \equiv P_{ik} P_{jl} - \frac{1}{2} P_{ij} P_{kl}$. The Lambda tensor $\Lambda_{ij,kl}$ is transverse in *all* indices, i.e, $n^i \Lambda_{ij,kl} = 0$, and it projects out the trace $\Lambda_{ii,kl} = \Lambda_{ij,kk} = 0$. By projecting the metric perturbation, we obtain the traceless transverse version

$$h_{ij}^{TT} = \Lambda_{ij,kl} \; h_{kl}. \tag{3.10}$$

As an example, for a wave in $\hat{z}$ direction, the projector will have the form

$$P = \begin{pmatrix} 1 & 0 & 0 \\ 0 & 1 & 0 \\ 0 & 0 & 0 \end{pmatrix}, \tag{3.11}$$

and any arbitrary symmetric matrix will take the form

$$\Lambda_{ij,kl} \underbrace{A^{kl}}_{\substack{\text{Arbitrary sym-}\\ \text{metric } 3 \times 3 \\ \text{matrix}}} = \begin{pmatrix} \overset{\displaystyle \nearrow h_+}{\frac{1}{2}(A_{11} - A_{22})} & \overset{\displaystyle \nearrow h_\times}{A_{12}} & 0 \\ A_{21} & -\frac{1}{2}(A_{11} - A_{22}) & 0 \\ 0 & 0 & 0 \end{pmatrix}.$$

To summarize, in the TT gauge, the metric perturbation is given by the quadrupole formula

$$h_{ij}^{TT}(t, \vec{x}) = \frac{2G}{c^4} \frac{1}{r} \Lambda_{ij,kl} \, \dddot{I}^{kl}(t - \frac{r}{c}), \tag{3.12}$$

where the quadrupole moment is defined as

$$I_{kl} \equiv \int d^3y (y_k y_l - \frac{1}{3} y^2 \delta_{kl}) T_{00} = I_{kl} - \frac{1}{3} I_m^m \delta_{kl}. \tag{3.13}$$

This is the trace-free version of $I$. It is a bit redundant with the projector $\Lambda$, although often useful in practice!

## 3.2 Energy momentum tensor of gravitational waves

So far, we have considered linearized Einstein equations as an expansion around the flat space-time metric $\eta_{\mu\nu}$. The fluctuations around the static flat background are the gravitational waves. In a general dynamical curved spacetime with a metric

$$g_{\mu\nu}(x) = \bar{g}_{\mu\nu}(x) + h_{\mu\nu}(x), \tag{3.14}$$

the question arises whether the curvature is actually a gravitational wave or part of the background [1]. In the latter case, the gravitational wave can locally be gauged away. How do we decide which part is the background and which part is a gravitational wave? A natural splitting arises when picking the right scale. Denoting the length of the background by $L_b$, and the wavelength of the gravitational wave by $\lambda_{GW}$.[5] A suitable length scale $d$ is large enough to observe $\lambda_{GW}$ and small enough such that the background is approximately flat. This method of separation of the metric into a smooth background and perturbations is called short-wave expansion.

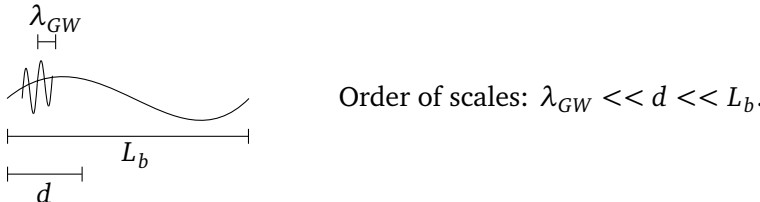

Order of scales: $\lambda_{GW} << d << L_b$.

Figure 4: *A visualization of short-wave expansion. The spacetime is separated in a background with a length $L_b$, and a gravitational wave with wavelength $\lambda_{GW}$. This separation arises natural when considering a scale length $d$, such that $\lambda_{GW} << d << L_b$.*

How does this perturbation propagate in the background spacetime and how does it affect the background metric [1]? To address these questions we expand the Einstein equations around a background metric. In this expansion there are typically two small parameters, the amplitude $h$ and $\frac{\lambda_{GW}}{L_b}$ (or $\frac{f_B}{f}$). So let us expand $G_{\mu\nu}$ in powers of $h$:

$$G_{\mu\nu} = G_{\mu\nu}^{(B)} + G_{\mu\nu}^{(1)} + G_{\mu\nu}^{(2)} + \dots \tag{3.15}$$

The term $G^{(B)}$ is related to the background and solely constructed from $\bar{g}_{\mu\nu}$. $G_{\mu\nu}^{(1)}$ is linear in $h_{\mu\nu}$ and contains only high-frequency modes, while $G_{\mu\nu}^{(2)}$ is quadratic and contains *both* high and low frequencies. For instance, consider a quadratic term $h_{\mu\nu}h_{\rho\sigma}$, where $h_{\mu\nu}$ and $h_{\rho\sigma}$ contain

---

[5]Note that the typical length scale for the gravitational wave is $\lambda_{GW} = \frac{\lambda}{2\pi}$ instead of $\lambda$ and is known as the reduced wavelength.

a mode with wave-vector $\vec{k}_1$ and $\vec{k}_2$, respectively, with $|\vec{k}_1|, |\vec{k}_2| >> \frac{1}{d}$. The high frequency wave vectors can combine such that the sum becomes a low frequency wave vector mode, $|\vec{k}_1 + \vec{k}_2| << \frac{1}{d}$. In this manner, the Einstein equations can be split into equations for high frequencies and for low frequencies. We will focus on the small $\vec{k}$ [6] part of Einstein's equation

$$
\begin{aligned}
G^B_{\mu\nu} &= -[G^{(2)}_{\mu\nu}]^{small\,\vec{k}} + \frac{8\pi G}{c^4}[T_{\mu\nu}]^{small\,\vec{k}} \\
&= -\left\langle G^{(2)}_{\mu\nu}\right\rangle_d + \frac{8\pi G}{c^4}[T_{\mu\nu}]_d
\end{aligned}
\tag{3.16}
$$

In the second line, we average over a spatial volume at a scale $d$. This does not affect the modes with a wavelength of order $L_b$, since these are more or less constant over a distance $d$. On the other hand, the fast oscillating waves of order $\lambda_{GW}$ will average to zero. The attentive observer will notice that the above technique is basically a renormalization group transformation. We take the fundamental equations of the theory and "integrate out" the small (high energy) fluctuations, to obtain an effective theory that describes physics at the length scale $L_b$. The result is the *course-grained* Einstein equations.

The averaged $2^{nd}$ order $G_{\mu\nu}$ is defined as the energy-momentum tensor of gravitational waves

$$
t_{\mu\nu} = -\frac{c^4}{8\pi G}\left\langle G^{(2)}_{\mu\nu}\right\rangle_d = -\frac{c^4}{8\pi G}\left\langle R^{(2)}_{\mu\nu} - \frac{1}{2}\bar{g}_{\mu\nu}R^{(2)}\right\rangle.
\tag{3.17}
$$

Thus, gravitational waves carry energy that curves the background, because of the way it enters in (3.16). An explicit computation of $G_{\mu\nu}$ to $2^{nd}$ order in the TT gauge will give

$$
R^{(2)}_{\mu\nu} = \cdots = \frac{1}{4}\partial_\mu h_{\alpha\beta}\partial_\nu h^{\alpha\beta} + 12\,\text{terms},
\tag{3.18}
$$

$$
\left\langle R^{(2)}_{\mu\nu}\right\rangle_d = -\frac{1}{4}\left\langle\partial_\mu h^{TT}_{\alpha\beta}\partial_\nu h^{TT\alpha\beta}\right\rangle, \quad \left\langle R^{(2)}\right\rangle = 0, \quad \text{and} \quad \left\langle R^{(1)}\right\rangle = 0.
\tag{3.19}
$$

The explicit expression for $t_{\mu\nu}$ is then found by substituting $R^{(2)}_{\mu\nu}$ into equation (3.17),

$$
t_{\mu\nu} = \frac{c^4}{32\pi G}\left\langle\partial_\mu h^{TT}_{\alpha\beta}\partial_\nu h^{\alpha\beta TT}\right\rangle.
\tag{3.20}
$$

Furthermore, the energy density of gravitational waves is defined as the 00 - component of the energy stress tensor, and is given by

$$
\rho_{GW} = t_{00} = \frac{c^4}{32\pi G}\left\langle\dot{h}^{TT}_{ij}\dot{h}^{ijTT}\right\rangle.
\tag{3.21}
$$

## 3.3 Einstein's quadrupole formula

Given the energy density of gravitational waves, the energy of the gravitational radiation in a volume $V$ is given by

$$
E_{GW} = \int_V d^3x\, t^{00}.
\tag{3.22}
$$

Demanding conservation of energy-momentum tensor, $\partial_\mu t^{\mu\nu} = 0$, implies that

$$
\int_V d^3x(\partial_0 t^{00} + \partial_i t^{i0}) = 0
\tag{3.23}
$$

---

[6] $\vec{k} = \frac{2\pi}{\lambda} = \frac{1}{\lambda_{GW}}$

and we can write

$$\frac{dE_{GW}}{c\,dt} = -\int_V d^3x\,\partial_i t^{0i} = -\int_S dA\,n_i\,t^{0i}, \tag{3.24}$$

where $n_i$ is the outer normal to the surface and $dA$ the surface element of the volume $V$. Now, let S be a spherical surface at a large distance $r$ from the source. For a spherical volume, the surface element is $dA = r^2 d\Omega$, and its normal is $\hat{n} = \hat{r}$. Then

$$\frac{dE_{GW}}{dt} = -\frac{r^2}{c}\int d\Omega\,t^{0r} = \frac{r^2}{c}\int d\Omega\,t^{00}. \tag{3.25}$$

Hence, we have

$$P_{GW} = \frac{dE_{GW}}{dt} = \frac{r^2 c^3}{32\pi G}\int d\Omega\left\langle \dot{h}_{ij}^{TT}\dot{h}^{ij\,TT}\right\rangle = \frac{G}{8\pi c^5}\int d\Omega\,\Lambda_{ij,kl}(\hat{n})\left\langle \dddot{I}^{ij}\dddot{I}^{kl}\right\rangle, \tag{3.26}$$

which is known as Einstein's quadrupole formula:

$$P_{GW} = \frac{G}{5c^5}\left\langle \dddot{I}_{ij}\dddot{I}_{ij}\right\rangle, \tag{3.27}$$

describing the power emitted by a source with tensor moment $I_{ij}$. This allows for example to compute the gravitational wave emitted by a black hole binary.

## 3.4 The power spectrum of tensor perturbations

It is advantageous to parameterize the GW radiation emitted by a source through the GW spectrum, given by

$$\Omega_{\text{GW}} = \frac{1}{\rho_c}\frac{d\rho_{\text{GW}}}{d\ln k}. \tag{3.28}$$

This dimensionless quantity tells how the gravitational wave energy density is distributed across the Fourier modes. The energy density is source-dependent and is given by Eq. 3.21. We can write the tensor $h_{ij}$ in the Fourier space through

$$h_{ij}(t,\vec{x}) = \sum_{a=+,\times}\int \frac{d^3k}{(2\pi)^3}h_a(\vec{k})e^{i k_\mu x^\mu}\hat{e}_{ij}^a(\hat{n}); \tag{3.29}$$

$$= \sum_{a=+,\times}\int \frac{d k}{(2\pi)}\int_{S^2} d\Omega_{\hat{n}}\,h_a(k)e^{i k_\mu x^\mu}\hat{e}_{ij}^a(\hat{n}). \tag{3.30}$$

In these expressions, we already assumed that there are only two propagating degrees of freedom, the $+$ and $\times$ modes.[7] The polarization tensors are given by $\hat{e}_{ij}^a$, which are the components of the polarization tensor that maps the Cartesian coordinates of the tensor $h_{ij}^{TT}$ to the polarization modes $+,\times$.

It is important to realize that the GW spectrum depends on the 2-point function of the time derivative of the tensor $h_{ij}$. In turn, this function depends on the 2-point correlation function of the Fourier modes, given by

$$\langle h_a(\vec{k}_1)h_b(\vec{k}_2)\rangle = \delta_{ab}(2\pi)^3\delta^3(\vec{k}_1+\vec{k}_2)\frac{2\pi^2}{k_1^3}\mathcal{P}_t(k_1), \tag{3.31}$$

---

[7]Here we emphasize that the Fourier modes $h_a(\vec{k})$ and $h_a(k)$ have different units. In our notation, $[h_{ij}] = M^0$, therefore $[h_a(\vec{k})] = M^{-3}$, while $[h_a(k)] = M^{-1}$. This observation is useful to understand the parametrizations of the power spectrum in the expression of the 2-point correlation function in Eqs. 3.31 and 3.32.

where $\mathcal{P}_t(k$ is the dimensionless *power spectrum of tensor perturbations*. By knowing the power spectrum of tensor perturbations, we fully characterize the GW spectrum. We give some example in the last chapters of these notes.

Alternatively, we can parameterize the problem with the *power spectral density $S_h(k)$*

$$\langle h_a(k_1)h_b(k_2)\rangle = \frac{1}{2}\delta_{ab}\frac{\delta(\Omega_1, \Omega_2)}{4\pi}(2\pi)\delta(k_1 - k_2)S_h(k_1). \tag{3.32}$$

The power spectral density is a dimensional quantity. In the GW literature, $S_h$ is often multiplied by a frequency factor,[8] giving rise to the *characteristic strain*, a dimensionless quantity,

$$h_c(f) = \sqrt{fS_h(f)}. \tag{3.33}$$

# 4 The stochastic gravitational wave background

In the last lectures, we generically described gravitational waves, identifying the two propagating degrees of freedom from general relativity. Now, we specialize our studies on *stochastic gravitational waves* by deriving the main properties and describing current searches.

## 4.1 The gravitational wave background

Stochastic gravitational wave backgrounds (SGWBs) are defined as the superposition of gravitational waves with different wave numbers $\vec{k}$ (both in magnitude and direction). They can have astrophysical or cosmological origins and typically are isotropic, unpolarized, and Gaussian.[9] It is then very similar to the cosmic microwave background (CMB) from the electromagnetic spectrum. However, SGWBs can allow us to reach stages where CMB cannot guide us since gravitational waves can travel freely through the hot plasma of the early universe, which was not transparent to photons.

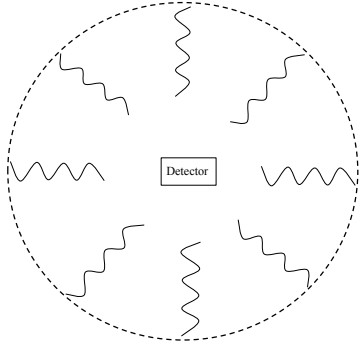

Figure 5: *Here we show a schematic representation of the propagation and detection of SGWBs. The circle represents some cosmic event (gravitational wave source). The waves then propagate through the universe. Occasionally, they find a detector. The signal from SGWBs acts as additional "noise" in a gravitational wave detector.*

Possible early-universe sources emitted gravitational waves in the past. These signals are continuously reaching us, coming from all directions. Signals from SGWBs are small, and it

---

[8]We can express these quantities in terms of the frequency mode $f$ by using the definition of the wavenumber length, $k = 2\pi(f/a_0)$.

[9]From astrophysical sources, this follows from the central limit theorem [3]. For cosmological sources, this statement is model-dependent.

is challenging to detect them because what arrives in the detector is similar to a noise source. Therefore, they can serve as a cosmological history book, which is tricky to decipher. Phenomena like inflation, primordial black holes, cosmic strings, and phase transitions are possible cosmic sources. These primordial sources are relevant for beyond-Standard-Model theories. These theories usually rely on energy scales far beyond what Earth-based experiments can achieve but were reached during the high-temperature stages of the early universe.[10]

Notice that SGWB signals are different than those LIGO-Virgo detected (transient signals) in the recent gravitational wave observation from binary mergers [4, 5]. These are transient signals produced by the merger of black-hole or neutron-star binaries. In these notes, we focus on the spectral properties and sources of the stochastic gravitational wave background. In the case of an astrophysical background [6], its nature is assumed to be continuous (because there is a large number of sources, the time interval between events is small compared to the duration of a single event and the background obeys Gaussian statistics [3]). We do not study shot noise (small number of sources) and popcorn (intermediate number of sources) processes, that might also be detected by ground-based interferometers [7].

Before June 2023, pulsar time array (PTA) collaborations had claimed statistical evidence for excess noise that early-universe gravitational waves could potentially explain, see, for instance, the reviews [8–10]. This excess noise was common red noise. In short, a common red noise process can be modeled with a frequency-dependent red-shifted spectrum across scales. Detection of GWs, however, depends on evidence for the Helling-Downs correlation [11]. At the end of June 2023, enormous attention was given to the latest data release of the PTA collaborations. We could watch exciting results from NANOGrav, CPTA, EPTA, and PPTA. Even though the number of $\sigma$s are below 5, for the first time, the collaborations found significant evidence for Helling-Downs [12–15], suggesting a GW origin of the signal. We comment more on PTA searches in the following sections.

PTAs can probe frequencies around one nHz. Beyond PTAs, gravitational-wave detectors are expected to reach the sensitivity required for SGWBs with LISA (in the frequency range around one mHz) [16] and the next generation of ground-based detectors. So far, the LIGO-Virgo-Kagra - LVK - collaboration put upper bounds on the background at LVK scales (around 100 Hz) [17].

To understand how these observatories can detect gravitational waves, we introduce the experiments, discuss the interferometer idea, and scrutinize PTA searches.

## 4.2  Experiments

Here, we briefly comment on three different experimental configurations that can probe the SGWB: i) ground-based interferometers, ii) space-born interferometers, and iii) pulsar timing arrays. Together, they cover different frequency ranges of the GW spectrum, see Fig. 6.

### 4.2.1  Ground-based interferometers

LIGO, Virgo, and KAGRA [18–20] (and their joint LVK collaboration) are collaborations based on ground-based interferometers. Their interferometers follow the idea behind the Michelson-Morley interferometers, see Sec. 4.3.1 and Fig. 7. So far, the LVK collaboration has detected only transient signals of GWs. Transient signals are well localized in time and space, as shown in the famous Fig. 1 of [4], in opposition to the SGWB. Their detectors are sensitive to probe the background at frequencies in the range 10–1000 Hz

---

[10]For instance, current accelerators reach center-of-mass energies of the order $\sim 10^4$ GeV, which is enough to probe the electroweak phase transition scale, $\sim 100$ GeV. Still, it is far below grand-unification scales of order $\sim 10^{16}$ GeV.

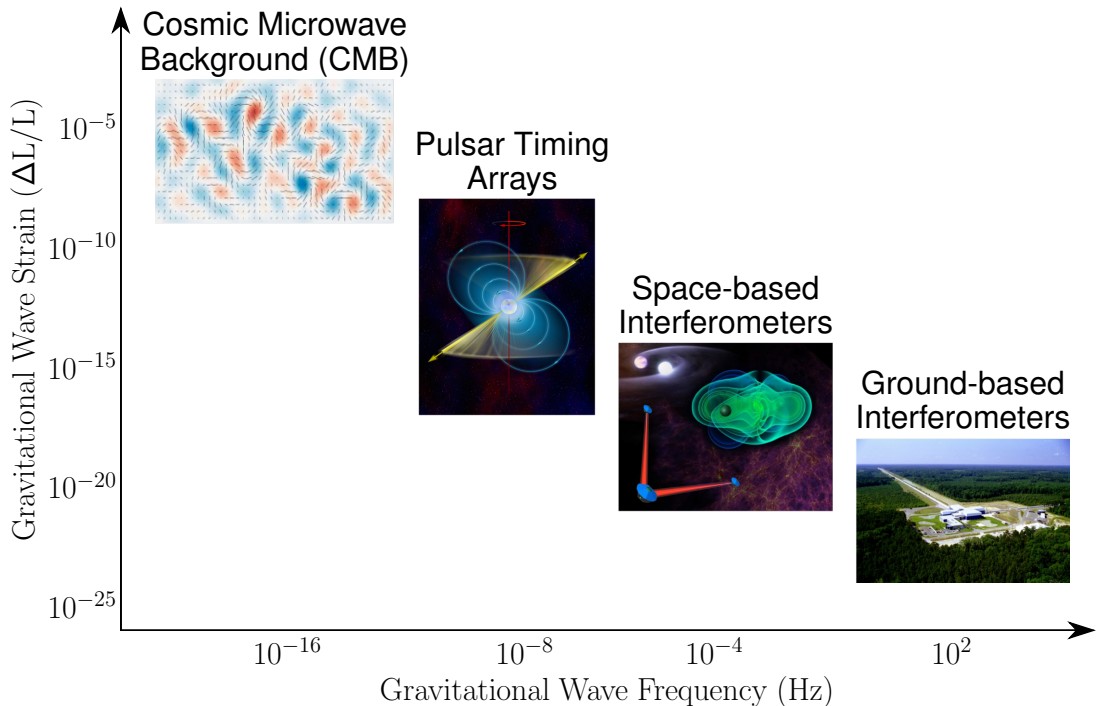

Figure 6: *Scheme of different experiments probing the gravitational wave spectrum. Thanks to Michael Lam from the NANOGrav collaboration for sharing the template. From left to right, the images are credited to BICEP2, Bill Saxton at NRAO, eLISA, and LIGO, respectively.*

As the gravitational wave perturbs the spacetime, it delays photon beams traveling across the interferometer's arms. In turn, the small perturbation disturbs the perfect interference pattern that would exist without GWs. The resulting gravitational wave strain is given by the ratio $\frac{\Delta L}{L} \sim |h_{\mu\nu}| \sim 10^{-21}$, where $L$ is the length of the arm and $(L \pm |\Delta L|)$ is the corresponding effective length in the presence of GWs. In the first GW detection in 2015 [4], the two pair of interferometers observed a transient GW signal whose *characteristic strain* was of the order $|h_{\mu\nu}| \sim 10^{-21}$, with arm lengths of $L \sim 10^3$ m, resulting in a very small $\Delta L \sim 10^{-18}$ m. Compared with the proton dimension $10^{-15}$ m, the gravitational wave interferometer must be extraordinarily sensitive!

These experiments rely on very large arm lengths, and their detectors possess other enhancements to achieve higher sensitivities. Consider, for instance, a signal whose peak frequency is at $f \sim 100$ Hz, typical for binary mergers, such that $\lambda \sim L \sim \frac{1}{2\pi f} \sim 750$ km. For such wavelength, construction would be impossible. The trick is to use resonant Fabry-Perot resonant optical cavities, which reduce the required size to $L \sim 4$ km, by making the laser beam bounce between the two mirrors around 300 times. Moreover, the characterization of the fringe patterns in the photodetector is strongly dependent on the laser power. The detectors of LIGO operated close to 750 kW to detect those GWs in 2015. Such power could be obtained with power recycling mirrors.

The combination of detectors improves the experimental power and sensitivity. Indeed, if one uses two detectors, one can analyze auto and cross-correlations: in the first gravitational wave detection by the Ligo-Virgo collaboration [4], there were two LIGO interferometers in the USA. For three detectors, one can analyze the 3D localization and polarization of

isotropic SGWB: in the first multi-messenger gravitational wave detection from neutron-star binary merger [5], there were three interferometers (two in the USA, one in Italy). Beyond LVK, the future generation of ground-based interferometers includes the Einstein telescope [21] and the Cosmic Explorer [22]. These experiments probe the astrophysical and cosmological background in the $10 - 10^3$ Hz range, probing cosmology, new physics, and fundamental physics, see, for instance, [7, 23–31].

Because of their arm lengths, the LVK collaboration probes frequencies in the $10 - 10^3$ Hz range. In turn, this range limits the mass of the black hole binaries they can probe. There are at least two ways to circumvent this problem: space-born interferometry, by building an interferometer in space with long arm lengths, such as LISA [32]; and PTA searches, by cross-correlating photon signals emitted by different pulsar – each pair pulsar-Earth can be understood as a galactic-sized interferometer arm – [33].

### 4.2.2   Space-born interferometers

In interferometers, the smaller the frequency, the larger the arm. This fact limits the sources that a detector can observe.

LISA [16] is the future space-based interferometer to be launched in the 2030s. The idea is the same as ground-based interferometers, but in space, the interferometer will follow Earth's orbit around the sun. LISA will have three $2.5 \times 10^6$ km arms that would probe frequencies in the mHz range, with three detectors able to detect two independent signal channels.

This experiment would probe smaller frequencies that, on Earth, would require unrealistic large arms. For such frequencies, it would be possible to detect ultra-compact binaries in our galaxy, supermassive black hole mergers, extreme mass ratio inspirals, and other plethora of cosmological possibilities associated with early universe physics [34–37], constituting LISA a promising probe of fundamental and new physics.

Moreover, the Japanese Deci-hertz Interferometer Gravitational Wave Observatory (DECIGO) is another space-born mission. They aim to probe GWs in the frequency band of 0.1 Hz to 10 Hz [38].

### 4.2.3   Pulsar timing arrays

Pulsar time array (PTA) searches do not work by the two interferometer strategies presented above. Instead, they rely on detecting gravitational waves by measuring the time of arrival of radio pulses from millisecond pulsars. These pulses are disturbed by spatially correlated fluctuations induced by gravitational waves.

As gravitational waves perturb the metric along the Earth-pulsar lines, they modify the time of arrival of the radio pulses on Earth. A set of PTAs creates correlations across the baselines, while other noise sources are uncorrelated. Searches using PTAs then compare the measured spatial correlations with the expected values from the Helling-Downs curve, the smoking gun for the isotropic, unpolarized background of quadrupole gravitational wave radiation [11, 33, 39]. NANOGrav [40], PPTA [41], EPTA [42], as well as their joint international consortium IPTA [43] are among the most extensive PTA collaborations. The collaborations recently claimed hints for detection on SGWB, see Sec. 4.4.[11]

PTAs can probe frequencies in the nHz range. These frequencies are lower than what ground- and space-based interferometers can observe. In this range, supermassive black hole binaries are the primary source of gravitational waves. Supermassive black holes have masses larger than $10^5$ solar masses, are present in the center of galaxies, and are much heavier than those producing the transient signals detected by the LVK collaboration. Since the frequency

---

[11]At the same frequency range, the Square Kilometre Array (SKA) collaboration [44] promises to increase the sensitivity further in the following years.

of gravitational waves scales as the inverse of the binary chirp mass, neither LVK nor LISA can detect such supermassive black holes.

On top of this astrophysical background, a cosmological background produced by early-universe and BSM physics can also be probed with PTAs [45].

For details on the past, present, and future of PTA collaborations, see, for instance, [33,46]. A recent and useful reference to compare the searches for the SGWB in interferometer and pulsar timing data is [47].

## 4.3 Searching for the background with interferometers

In this section, we describe how we can detect GWs using Michelson interferometers. This is the technique used by LIGO-Virgo in the first detections. The experiment consists of probing the interference pattern perturbed by gravitational waves. We comment on PTA searches in Sec. 4.4.

### 4.3.1 Experimental setting

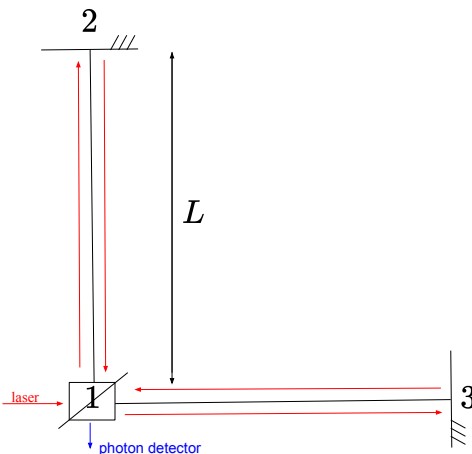

Figure 7: *Michelson interferometer: the experiment consists in splitting laser light beams through a beam splitter at 1 in two different paths (here each arm has length L), re-flecting them with two mirrors located at 2 and 3, and recombining them back in 1 so that interference patterns can be created (in particular, perfect destructive interference) in the photon detector (dark fringe).*

According to the setting in Fig. 7, we need to compute the time delay associated with light departing and returning to 1. There are at least two possible frames: in the TT frame, gravitational waves change the photon propagation, and the "free-falling" mirrors do not move; in the proper detector frame, gravitational waves change the distance between the beam-splitter and mirrors.

Let us work in the TT frame. If light travels with $c = 1$, it takes $L$ to travel from 1 to the mirror and $2L$ back to 1. In the $\hat{l} = \vec{x}$ direction, the time delay for a light signal emitted at time $t$ is given by:[12]

$$\Delta T(t) = \frac{1}{2}\hat{l}^a\hat{l}^b \int_0^L h_{ab}(t+s, \vec{x}+s\hat{l})ds, \qquad a,b = 1,2,3; \qquad (4.1)$$

---

[12]Here, it suffices to use flat spacetime. There is no relevant back-reaction of the gravitational waves in the spacetime where they are propagating, according to Eq. 3.

where

$$h_{ab}(t,\vec{x}) = \int d^3k \, e^{-2\pi i \vec{k}\cdot\vec{x}} \sum_\lambda \hat{e}_{ab,\lambda}(\hat{k}) h_\lambda(t,\vec{k}). \tag{4.2}$$

The time delay, to be measured at time $t$, associated with the return trip is $\Delta T_{12}(t-2L) + \Delta T_{21}(t-L)$. We also expect to find some noise $n_1(t)$. Thus, we write

$$\begin{aligned} s_{12}(t) &= \Delta T_{12}(t-2L) + \Delta T_{21}(t-L) + n_1(t) & (4.3) \\ &= L \int \frac{d^3k}{(2\pi)^3} \sum_\lambda I_\lambda^{12}(\vec{k}) h_\lambda(t-L,\vec{k}) + n_1(t), & (4.4) \end{aligned}$$

where we have defined the single-arm detector response as

$$I_\lambda^{12}(\vec{k}) = \frac{1}{2}\hat{l}^a\hat{l}^b\hat{e}_{ab,\lambda}(\hat{k})e^{-2\pi i\vec{k}\cdot\vec{x}} \times$$
$$\times \left( e^{-\pi ikL(1+\hat{k}\cdot\hat{l})}\frac{\sin(\pi kL(1-\hat{k}\cdot\hat{l}))}{\pi kL(1-\hat{k}\cdot\hat{l})} + e^{\pi ikL(1-\hat{k}\cdot\hat{l})}\frac{\sin(\pi kL(1+\hat{k}\cdot\hat{l}))}{\pi kL(1+\hat{k}\cdot\hat{l})} \right). \tag{4.5}$$

Notice that this expression tells us the direction in the detector is more sensitive! It also tells us something about frequency modes: the terms in the last bracket tend to be 2 for $kL << 1$ (small-frequency modes) and to 0 for $kL >> 1$ (large-frequency modes). From the response function, there is suppression for both small and large frequency modes. For the large ones, the signal drops as $(\sin x)/x$ for $k >> 1/L$. For the small ones, the response function is constant for $k << 1/L$. As the noise grows at low frequencies, sensitivity is lost.

### 4.3.2 Overlap reduction function

Assuming isotropic SGWB, the measured time delay $s_\alpha$ can be averaged

$$<s_\alpha^2> = L^2 \int \frac{d^3k}{(2\pi)^3} \sum_\lambda P_\lambda(k) \, | \, I_\lambda^{12} - I_\lambda^{13} \, |^2 + <n^2>, \tag{4.6}$$

where $P_\lambda$ is the non-normalized isotropic power spectrum

$$\langle h_\lambda(\vec{k}_1) h_{\lambda'}(\vec{k}_2) \rangle = (2\pi)^3 \delta_{\lambda\lambda'} \delta^3(\vec{k}_1 - \vec{k}_2) P_\lambda(| \, \vec{k}_1 \, |), \tag{4.7}$$

whose normalized version was introduced in Eq. 3.31. Above, $I_\lambda^{12} - I_\lambda^{13} = I_\lambda^\alpha$, and $<n^2>$ is the instrumental noise from different possible sources. $P_\lambda$ depends on the gravitational wave signal and $I_\lambda^\alpha$ on the detector response.

In such cases, the main obstacle is noise. Consider that noise severely affects the data for very tiny time delays (which we expect from stochastic gravitational waves). Now, assume that there are two interferometers $\alpha$ and $\beta$, as shown in Fig. 8. Then,

$$<s_\alpha s_\beta> = L^2 \int \frac{d^3k}{(2\pi)^3} \sum_\lambda P_\lambda(k) I_\lambda^{\alpha*}(\vec{k},\vec{x}_1) I_\lambda^\beta(\vec{k},\vec{x}_2), \tag{4.8}$$

i.e., there is cross-correlation. Ideally, the two detectors are far away, so their instrumental noises are not correlated $<n_\alpha n_\beta> = 0$. The signal is reduced by *overlap reduction function*,[13] which is essential for the detection of stochastic signals.

---

[13]Although we get rid of the noise by considering two detectors, the measured time delay now depends on the distance between them, see Fig. 8. The overlap reduction function, therefore, depends on the response of the individual detectors and their relative geometry. For pulsar time arrays, the overlap reduction function is known as the *Hellings-Downs curve* [48].

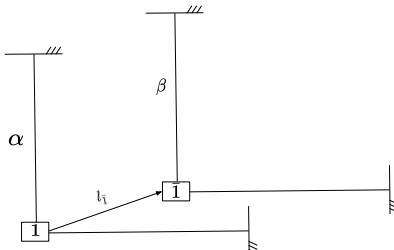

Figure 8: *Two Michelson interferometers. We put the origin of the coordinate system at 1. To describe the location of the second detector, we need to include the vector from 1 to $\bar{1}$, where $\bar{1}$ is the central point of the second detector. Therefore, $< s_\alpha s_\beta >$ depends on the distance between the interferometers.*

### 4.3.3  Monopole response function

Consider an isotropic, unpolarized SGWB, for which $P_+ = P_\times$. For this case, we can define the monopole response function as

$$\mathcal{M}(k) \equiv \sum_\lambda \int d\Omega \, | \, I_\lambda^\alpha \, |^2 \, . \tag{4.9}$$

Then, the average time delay is

$$\frac{< s_\alpha^2 >}{L^2} = \frac{1}{8\pi} \int d\,k \sum_\lambda \left( k^2 P_\lambda(k) \right) \left( \int d\Omega \, | \, I_\lambda^\alpha \, |^2 \right) = \frac{1}{4\pi} \int d(\ln k) \left( k^3 P_+(k) \right) \mathcal{M}(k). \tag{4.10}$$

The dependence on the source is due to $P_\lambda(k)$, which is proportional to the amplitude of tensor perturbation, $\Delta_t^2$ (see Eq. 5.38, where we study the power spectrum of tensor fluctuations). The dependence on the configuration of the detector is due to $\mathcal{M}(k)$. Then, by measuring the time delay and knowing the configuration of the detector, it is possible to probe the source!

For instance, in the LIGO-Virgo detectors, two detector arms are oriented perpendicularly to each other. The sensitivity and the monopole response function depend on the wave number, as shown in Fig. 9.

For polarized sources, $P_\times(k) \neq P_+(k)$, we should repeat the computation and keep $\langle hh \rangle$ in $\sum_\lambda$. One can search for polarization through the added multiples. For non-isotropic sources, we should repeat the computation and keep $\langle hh \rangle$ in the integral over $d\Omega$. One can search for anisotropies through antenna patterns, see Fig. 10.

### 4.4  PTA searches

The idea behind PTA searches is to use millisecond pulsars at the nearby galactic environment and their precise time-dependence to develop a galaxy-size interferometer, where each pair pulsar-Earth can be seen as a super long arm in a big Michelson interferometer. Photons are emitted by pulsars, which are very regular lighthouses across the sky [49–53], and the GW interference to these photons redshifts the pulse that arrives in a radio telescope on Earth. As described in Sec. 4.3.2, cross-correlation helps to distinguish a GW signal from other noise sources. In the PTA context, this means that signals of multiple pulsars must be observed. Huge radio telescopes on Earth have observed these pulsars for several years. Then, by cross-correlating the accumulated timing observation and designing precise noise models, we can investigate whether the overlap reduction function is consistent with the Hellings-Downs correlation [11].

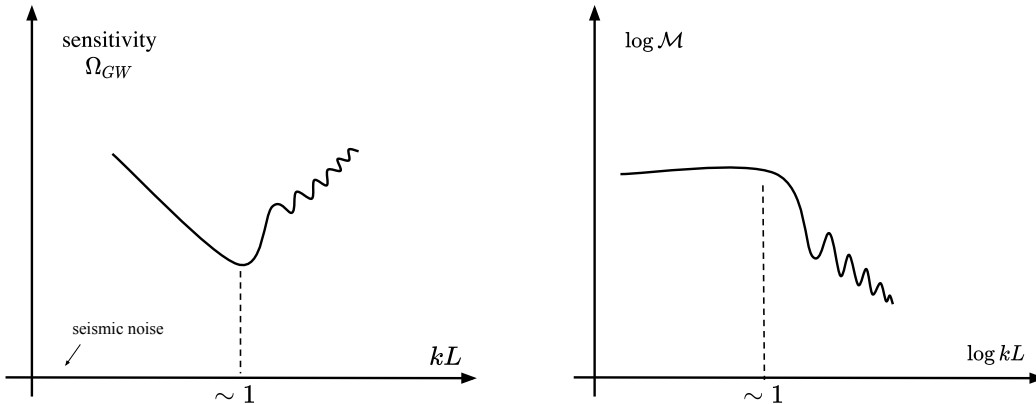

Figure 9: *On left, sensitivity of gravitational waves versus kL (length of detector L), given dependence of signal on the correlation $< s_\alpha^2 >$ through Eq. 4.10. For small wavelengths, seismic noise is an obstacle. For large wavelengths, the monopole response oscillates too much. On the right, log-log plot of the monopole detector response versus kL. The detector response drops for $kL >> 1$ (averaging over many oscillations, see Eq. 4.9), and it is constant for $kL << 1$.*

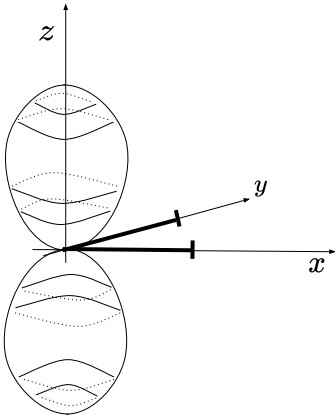

Figure 10: *The interferometer and detector are located at the $xy$ plane. The antenna pattern is a detector property and is useful for detecting anisotropy.*

The frequency of the GW that can be observed with PTA searches is constrained by how long observations are performed – the minimal accessible frequency is $f_{\min} \sim 1/T \sim 10^{-9}$ Hz, for $T \sim 10$ years of total observing time – and how small the observing cadence is – the maximal accessible frequency is the observation window in each observatory $f_{\max} \sim 1/T \sim 10^{-6}$ Hz, if the telescope can be used every other week.

Importantly, every pulsar has a *timing model* that accounts for every known influence on the photon propagation between the pulsar and the radio telescope. The timing model contains timing corrections associated with corrections within the Solar System, in the pulsar systems, and dispersions [33, 54–56]. The difference between the time of arrival and the predictions of the timing model for a pulsar gives the *timing residual* of that pulsar. Everything that is not included in the timing model accounts for timing residuals. This includes the desired signal of GW and also other sources of noise. In turn, these noise sources are modeled as Gaussian processes in *noise models* [33, 57–59]. Noise sources can usually be classified as *white noise* or *red noise* processes. The latter is modeled by a frequency-decreasing power spectrum, whereas the former is modeled by a constant power spectrum over the frequency domain.

Finally, the search for a GWB requires a joint estimation of the GW and noise signals [33]. It proceeds primarily with a Bayesian inference analysis that relies on model comparison and it can be complemented by frequentist detection statistic techniques (the *optimal statistic* [60–62]), in which a null hypothesis testing is obtained by averaging over the Bayesian posteriors of the noise parameters [63]. See more details in [12,64].

In the past few years, we have been experiencing considerable advances in PTA searches for the GWB. These advances are colossal technological and intellectual achievements. It is a technological achievement because a realistic search requires huge radio telescopes and enormous computational power. And it is also very challenging to reach the very high required precision to time and correlate the signals amid so much noise. The trailblazing idea behind finding evidence of the GWB within PTA searches and all the timing and noise modelings is, therefore, an intellectual achievement.

A detection would confirm one more prediction of GR, this time encoded in the Hellings-Downs correlation [11]. The expected GW source in the nHz frequency range is an astrophysical background composed of many supermassive black-hole binary (SMBHB) mergers. Supermassive black holes (SMBHs) are very massive astrophysical objects that populate the center of galaxies. The direct detection of gravitational waves from SMBHBs has never occurred since the binaries detected by LIGO-Virgo and LVK are composed of less massive black holes. In contrast, PTAs search for GWs at the right frequency band to detect these astrophysical signals.

Another source of SGWB is associated with primordial gravitational waves, as different sources in the early universe could have emitted these gravitational waves. This is relevant because we can probe inflationary cosmology and BSM physics.

Together with LVK and the future LISA space interferometer [65], PTA searches are, therefore, part of an extensive program of probing fundamental physics with gravitational waves. In the following section, closely following [33, 66], we derive the correlation function of the timing residuals of a pair of pulsars, which allows us to obtain the Helling-Downs correlation.

### 4.4.1 Timing residuals and the Hellings-Downs correlations

We define the residual of the pulse arrival time for a single pulsar as

$$R(t) = \int_0^t d\tilde{t} \left( \frac{\nu_0 - \nu(\tilde{t})}{\nu_0} \right) = \int_0^t d\tilde{t}\, z(\tilde{t}). \tag{4.11}$$

The frequency in flat spacetime is $\nu_0$, and the redshift at a given time is $z(t)$. The derivation of the redshift can be found in [33]. It depends on $\hat{p}$, a unit vector pointing from Earth to the pulsar, and on the difference between tensor modes of the metric perturbation, $h_{ij}$, at the pulsar and on Earth, for a GW traveling along a direction $\hat{\Omega}$. The GWB consists of waves from all directions. Therefore, the total redshift of a single pulsar in Fourier space is obtained after integrating over all directions,

$$\tilde{z}(f,\hat{p}) = \int d^2\Omega\, z(f,\hat{p},\hat{\Omega}) = \left( e^{-2\pi i f L(1+\hat{\Omega}\cdot\hat{p})} - 1 \right) \sum_a h_a(f,\hat{\Omega}) F^a(\hat{\Omega}), \tag{4.12}$$

where

$$F^a(\hat{\Omega}) = \frac{1}{2} \frac{\hat{p}_i \hat{p}_j}{(1+\hat{\Omega}\cdot\hat{p})} \hat{e}_{ij}^a(\hat{\Omega}) \tag{4.13}$$

is the $a^{\text{th}}$-GW *antenna response pattern* mode for the pair Earth-pulsar.

With these expressions, we can compute the residual $R(t)$,

$$R(t) = \frac{1}{2} \frac{\hat{p}_i \hat{p}_j}{(1 + \hat{\Omega} \cdot \hat{p})} \int_0^t d\tilde{t} \Delta h_{ab}, \tag{4.14}$$

where

$$\Delta h_{ab} = \left[ h_{ab}(t, \mathbf{x}_{\text{Earth}}) - h_{ab}(t - L, \mathbf{x}_{\text{pulsar}}) \right] = \int_{-\infty}^{\infty} df \, \tilde{z}(f, \hat{p}). \tag{4.15}$$

Next, we add a second pulsar to compute the cross-correlation of a pair of pulsars. Assuming a Gaussian, stationary, unpolarized, spatially homogeneous, and isotropic background, according to Eqs. 3.32, 4.11, and 4.12, the correlation between the two pulsars is

$$\langle R(t_1, \hat{p}_1) R(t_2, \hat{p}_2) \rangle = \int_{-\infty}^{\infty} df \, e^{2\pi i f(t_1 - t_2)} S_h(f) \Gamma(f, \xi), \tag{4.16}$$

for an angular separation between two pulsars $\xi = \cos^{-1}(\hat{p}_1 \cdot \hat{p}_2)$. In this expression, there are two important quantities. While $S_h(f)$ encodes information about the source of the background – as it is related to the power spectrum of the tensor perturbations, see Eq. 3.32 – the quantity $\Gamma(f, \xi)$ is the *overlap reduction function*, which contains information about the geometry of the system. This quantity is analogous to the overlap response function in Sec. 4.3.2. In particular, it tells how the GW radiation power is distributed along the sky according to the angular distribution of pulsars. It is the fingerprint of an SGWB, known as Hellings-Downs correlation [11].

To obtain this function, we use the following correlation of redshifts

$$\langle \tilde{z}_1(f_1, \hat{p}_1), \tilde{z}_2^*(f_2, \hat{p}_2) \rangle = \frac{C_1}{2} \delta(f_1 - f_2) S_z(f)_{12}, \tag{4.17}$$

where $S_z(f)_{12}$ is the cross-power spectral density of the measured redshifts of the signals from pulsars 1 and 2. $C_1$ is a normalization factor that is fixed below. By explicitly computing the left-hand side of this correlator with Eq. 4.12, we derive an expression for this spectral density:

$$S_z(f)_{12} = \frac{1}{2} H(f) \int \frac{d^2\Omega}{4\pi} \left( e^{-2\pi i f L_1(1 + \hat{\Omega} \cdot \hat{p}_1)} - 1 \right) \left( e^{2\pi i f L_2(1 + \hat{\Omega} \cdot \hat{p}_2)} - 1 \right) \sum_a F_1^a(\hat{\Omega}) F_2^a(\hat{\Omega}). \tag{4.18}$$

Back to Eq. 4.16, we integrate it over time and use the definitions in Eqs. 4.11 and 4.17, to find an expression for the overlap reduction function by comparison with Eq. 4.18.

All pulsars probed by the PTA collaborations are millisecond pulsars that belong to our cosmic galactic neighborhood. They are at least 100 parsecs (about $3 \times 10^{18}$ m) distant from us, while the typical frequency values are of order $10^{-9}$ Hz. Therefore, $fL \gg 10$, and the dependence on the frequency can be neglected [67]. Consequently, the brackets in Eq. 4.18 are 1 (for distinct pulsars) or 2 (for identical pulsars, i.e., same $L$, $\hat{\Omega}$, and $\hat{p}$). The overlap reduction function reduces to $\Gamma(f, \xi) \to \Gamma(\xi)$,

$$\Gamma(\xi) = C_1 \int \frac{d^2\Omega}{4\pi} \sum_a F_1^a(\hat{\Omega}) F_2^a(\hat{\Omega}). \tag{4.19}$$

We use the antenna pattern in Eq. 4.13 to write $\Gamma(\xi)$ as a function of the angular separation.[14] For a pair of pulsars $i$ and $j$ whose angular separation is $\xi_{ij}$, we write the *Helling-Downs* curve as

$$\Gamma_{\text{HD}}(\xi)_{12} = \frac{C_1}{12} \left( 4 + (\cos \xi - 1) + 6(1 - \cos \xi) \ln \frac{1 - \cos \xi}{2} \right). \tag{4.20}$$

---

[14]See an explicit parametrization of the angular dependence in $\hat{p}_i$ that defines $F_i^a$, for instance, in [68]. To simplify, we can define $\hat{p}_1 = \hat{z}$ and $\hat{p}_2 = \sin \gamma \hat{x} + \cos \gamma \hat{z}$ so that $F_1^\times = 0$. Therefore, the overlap reduction function only depends on the average of $F_1^+ F_2^+$ over the angular separation.

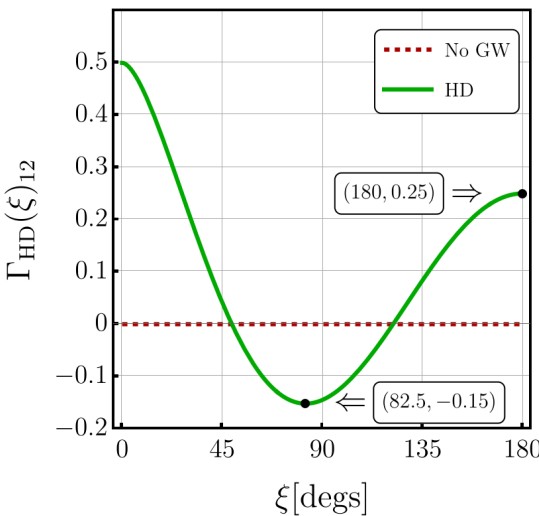

Figure 11: We show the normalized Helling-Downs curve – overlap reduction function for an isotropic SGWB in PTA searches – for two different pulsars as a function of the pulsar separation angle $\xi$. We also show the minimum and the value at 180 degrees.

If we work with identical pulsars, all terms proportional to $(1 - \cos \xi)$ vanish, while the remaining numerical factor is multiplied by a factor of two. Therefore, we can write

$$\Gamma_{\text{HD}}(\xi)_{ij} = \frac{C_1}{12}\left(4(1 + \delta_{ij}) + (\cos \xi - 1) + 6(1 - \cos \xi)\ln \frac{1 - \cos \xi}{2}\right). \qquad (4.21)$$

Next, we define $x_{ij} = (1 - \cos \xi_{ij})/2$. A usual convention is to set $\Gamma(0) = 1/2$ for distinct pulsars on the same line of sight and $\Gamma(0) = 1$ for identical pulsars. This fixes the normalization factor, $C_1 = 3/2$. In this case, we have $S_z(f)_{ij} = (1/3)H(f)\Gamma_{\text{HD},ij}$, and the Hellings-Downs correlation is

$$\Gamma_{\text{HD},ij} = \frac{1}{2}(1 + \delta_{ij}) - \frac{1}{4}x_{ij} + \frac{3}{2}x_{ij}\ln x_{ij}. \qquad (4.22)$$

This is the famous fingerprint that should be hidden in pulsar timing data if the signal comprises an isotropic GWB. We emphasize that the presence of the Hellings-Downs correlation in the data is independent of the source of the GWB, as the source contributes to $H(f)$. We plot the corresponding curve for $i \neq j$ as a function of the pulsar separation angle in Fig. 11. Since the function depends on the cosine of the angle, it is symmetric under $\xi \to 2\pi - \xi$.

Moreover, to understand how the Hellings-Downs correlation depends on multipolar modes, we can expand it in a Legendre polynomial basis

$$\Gamma_{\text{HD},ij} = \sum_{l=0}^{\infty} a_l P_l(\cos \xi_{ij}) = \sum_{l=2}^{\infty} \frac{3(l-2)!}{2(l+2)!}(2l+1)P_l(\cos \xi_{ij}). \qquad (4.23)$$

The main contribution is *quadrupolar* ($l = 2$, $a_2/a_3 \sim 25/7$), and there are no monopole ($a_0 = 0$) or dipole ($a_1 = 0$) contributions, as expected for GWs in GR.

Finally, we comment that throughout this section, we assumed that GR holds. Instead, we could also derive an expression for the overlap reduction function in models beyond GR, see, for instance, [69–71]. In these models, we not only have the two GR degrees of freedom because modified theories of gravity contain more propagating degrees of freedom.

We also assumed isotropy. More realistic astrophysical scenarios and even some cosmological models introduce anisotropies. These anisotropies would induce small perturbations to

the signal and modify the overlap reduction function. It is expected that anisotropies should exist and contribute very little to the signal, as in the CMB radiation [72–76].

### 4.4.2 The latest PTA data releases: HD on the radar!

It is impossible to finish the discussion on PTA searches without commenting on the latest results concerning the detection of Hellings-Downs correlations.

On June, 29th 2023, the main PTA collaborations released their latest datasets and a set of new papers exploring the data. A complete set of papers released that day by these collaborations are: by NANOGrav [12, 45, 56, 59, 64, 75, 77, 78], by EPTA (with InPTA data) [13, 79–83], by PPTA [14, 84, 85], and by CPTA [15]. These papers created considerable excitement because evidence for HD spatial correlations has been found for the first time [12–15], suggesting a GW origin of the signal.

Even though the collaborations obtained different confidence levels for HD and used different methods to analyze their data, they all agree that the latest datasets favor HD correlations over a model without HD. The NANOGrav collaboration, for instance, claimed evidence at 3–4 $\sigma$ level. The current results are highly encouraging to PTA searches and GW multimessenger astronomy. It is expected that evidence further increases in the next IPTA's data release 3. So far, see [86] for preliminary results in a joint analysis.

Besides searches for the HD correlation in the pulsar data, the collaborations also searched for signals of SMBHBs [77, 78, 82], new physics [45, 82, 83], and anisotropies [75].[15] At this moment, it is not possible to tell the source of the observed GWB. The interpretation of the origin of the signal is not conclusive because we still need to understand better how to model the astrophysical source (SMBHBs). The simplest scenario is modeled with a power-law power spectral density $S_h \sim f^{-\gamma}$, where $\gamma = 13/3$ is a constant. The resulting GW spectrum $\Omega_{\text{GW}} \sim f^{5-\gamma}$ is too much red-tilted to fit the pulsar data, explaining why many new physics models explained the data better in [45, 87]. These papers exemplify how powerful PTA searches are when it comes to probing fundamental physics.

## 5 Primordial gravitational waves

In Secs. 3 and 4, we saw how to compute the energy density $\rho_{\text{GW}}$ of gravitational waves and their spectrum $\Omega_{\text{GW}}$, and introduced interferometers and PTA searches. In this section, we learn how gravitational wave detectors can probe fundamental physics and different stages of the universe. First, we review some concepts in cosmology, then we establish some notions regarding the characteristic frequency of primordial GWs and we explore some properties of gravitational waves in an expanding universe. Finally, we comment on bounds from CMB and BBN.

### 5.1 A very short review on cosmology

At very large length scales, our universe is isotropic and homogeneous [88]. Comoving observers can describe this universe through the Friedmann-Robertson-Lemaître-Walker (FRLW) metric,

$$ds^2 = -dt^2 + a^2(t)\left(\frac{dr^2}{1 - \kappa r^2} + r^2 d\Omega^2\right), \tag{5.1}$$

in which $\Omega$ is the 3D solid angle, and $a(t)$ is the scale factor. The quantity $\kappa$ is a constant describing the curvature of the universe, assuming the values $+1$ if the universe is closed and

---

[15]Many follow-up papers exploring the data also appeared. We do not cite them here because they are many, and we would easily forget one or the other.

has positive curvature, 0 if the universe is open and flat, and $-1$ if the universe is open and has negative curvature (open). We can confidently set $\kappa = 0$ and work with a flat universe.

Next, we define the Hubble parameter is defined as

$$H(t) \equiv \frac{\dot{a}}{a}. \tag{5.2}$$

When we substitute the FRLW metric into the Einstein equations, we derive the Friedmann equations

$$H^2 = \left(\frac{\dot{a}}{a}\right)^2 = \frac{8\pi G \rho}{3}, \tag{5.3}$$

$$\dot{H} + H^2 = \left(\frac{\ddot{a}}{a}\right) = -\frac{1}{6}(\rho + 3p), \tag{5.4}$$

where $\rho$ is the total energy density, and $p$ is the momentum (given by the diagonal entries of the energy-momentum tensor), for a perfect fluid given by $T^{\mu}{}_{\nu} = \text{diag}(\rho, -p, -p, -p)$.

### 5.1.1 Properties of the FRLW solution

We can rewrite the Friedmann equations as

$$\frac{d\rho}{dt} + 3H(\rho + p) = 0. \tag{5.5}$$

With an equation of state $p = w\rho$, its solution is $\rho \propto a^{-3(1+w)}$. By substituting this result above, we find a solution for the scale factor

$$a(t) \propto t^{\frac{2}{3}(1+w)}, \quad w \neq -1, \tag{5.6}$$

and $a(t) \propto e^{Ht}$, $w = -1$ for a constant $H$. Therefore, matter ($w = 0$), radiation ($w = 1/3$), and any other type of fluid for which $w > -1/3$ do not produce accelerated expansion. Since the supernovae observation in the '90s, we know that the universe is accelerating [89, 90] so that we need to invoke some dark energy budget ($w \approx -1$). Similarly, in inflationary cosmology, the universe should have undergone a period of very fast expansion so that we can explain why the universe looks so homogeneous today, without problems of causality and fine-tuning of initial conditions in the past.

Finally, we also define the

$$\rho = \sum_i \rho_i, \quad \Rightarrow \quad \Omega_i = \frac{\rho_i^0}{\rho_c}, \tag{5.7}$$

where $\rho_c = 3H_0^2/(8\pi G)$ is the critical energy density, and $\Omega_i$ describes the fraction of each component making up the budget of our universe. According to [88], the FRLW describes the observable universe with $w_\Lambda \simeq -1$, $\Omega_\Lambda = 0.68$ (dark energy, cosmological constant), $\Omega_{\text{dm}} = 0.27$ (dark matter), $\Omega_b = 0.04$ (baryonic matter), and $H_0 = 67.4$ km s$^{-1}$ Mpc$^{-1}$.[16]

In this notes, we use $H$ to define scales. The size of the observable universe is set by the comoving particle horizon,

$$\tau \equiv \int_0^t \frac{dt'}{a(t')} = \int_0^{a(t)} \frac{da}{Ha^2} = \int_0^{a(t)} d\ln(a)\left(\frac{1}{aH}\right), \tag{5.8}$$

---

[16]A very interesting direction in cosmology is understanding the $H_0$ and $S_8$ tensions between values obtained with early-universe probes, such as the CMB [88], and late-time measurements, such as supernovae [91]. See, for instance, more details in [92, 93].

which is the light-like distance traveled by light from 0 to $t$ (the maximum distance an observer can travel). The Hubble horizon, $(aH)^{-1}$, describes the radius of a 2-sphere such that everything beyond its radius cannot be causally connected to the interior of the 2-sphere. Since $(aH)^{-1} = (1/H_0)a^{(1+3w)/2}$, the Hubble horizon increases over time for matter and radiation but decreases for standard slow-roll cosmic inflation, in which $H$ is approximately constant and $a$ increases. This notion of the Hubble horizon is fundamental to understanding the production of primordial gravitational waves in Sec 5.4. We comment more about this in the next section.

### 5.1.2   Cosmic inflation

For simplicity, we set here $M_p = 1, c = 1$. From the Einstein equations,

$$R_{\mu\nu} - \frac{1}{2}Rg_{\mu\nu} = T_{\mu\nu}, \tag{5.9}$$

for a FRLW spacetime, the Friedmann equations are given by Eqs. 5.3 and 5.4. We have decelerating expansion $\ddot{a} < 0$ if the equation of state parameter is $w = p/\rho > -1/3$. There are two problems associated with: the horizon and flatness problems.[17] Therefore, the condition $w < -1/3$ must be satisfied by a cosmic inflation model, that took place in the universe before BBN and radiation-domination era.

Consider a *single-field slow-roll* model

$$S = \int d^4x \sqrt{-g}\left(\frac{R}{2} - \frac{1}{2}g^{\mu\nu}\partial_\mu\phi\partial_\nu\phi - V(\phi)\right). \tag{5.10}$$

The energy-momentum tensor is given by[18]

$$T^{(\phi)}_{\mu\nu} = -\frac{2}{\sqrt{-g}}\frac{\delta S_\phi}{\delta g^{\mu\nu}} = \partial_\mu\phi\partial_\nu\phi - g_{\mu\nu}\left(\frac{1}{2}\partial_\alpha\phi\partial^\alpha\phi + V(\phi)\right). \tag{5.11}$$

By assuming that the field is homogeneous, i.e. $\phi(x,t) = \phi(t), \partial_i\phi = 0$,[19] we have

$$\rho_\phi = \frac{\dot{\phi}^2}{2} + V(\phi), \qquad p_\phi = \frac{\dot{\phi}^2}{2} - V(\phi), \tag{5.12}$$

and then

$$w_\phi = \frac{p_\phi}{\rho_\phi} = \frac{\dot{\phi}^2/2 - V(\phi)}{\dot{\phi}^2/2 + V(\phi)} \qquad \Rightarrow \qquad w_\phi \to -1 \quad \text{if} \quad V(\phi) >> \dot{\phi}^2/2. \tag{5.13}$$

The last condition is known as slow-roll, see Fig. 12, and therefore this model is able to describe a period of accelerated expansion, since $w_\phi < -1/3$. In particular, from the Friedmann equations, its solution is quasi de-Sitter, i.e. an exponential expansion $a(t) \propto e^{Ht}$, for a constant positive Hubble parameter $H$, that ends by the end of inflation

---

[17]The horizon problem is due to the fact that the Hubble horizon grows faster than any other physical scale so that the observed CMB spectrum implies uniformity for regions that were not causally connected in the early universe. The flatness problem is about the fact that the curvature of the universe is very small today, requiring even smaller curvatures in the past and very fine-tuned initial conditions. For more details on these problems, see, for instance, the lecture notes on inflation [94].

[18]Our convention is $g_{\mu\nu} = \bar{g}_{\mu\nu} + h_{\mu\nu}$, where $\bar{g}_{\mu\nu} = \text{diag}(-1, a, a, a)$.

[19]At the classical level, this assumption is required to satisfy the symmetries of the FRLW spacetime (isotropy and homogeneity). Quantum perturbations can then introduce anisotropies and inhomogeneities. We do not cover these cases here.

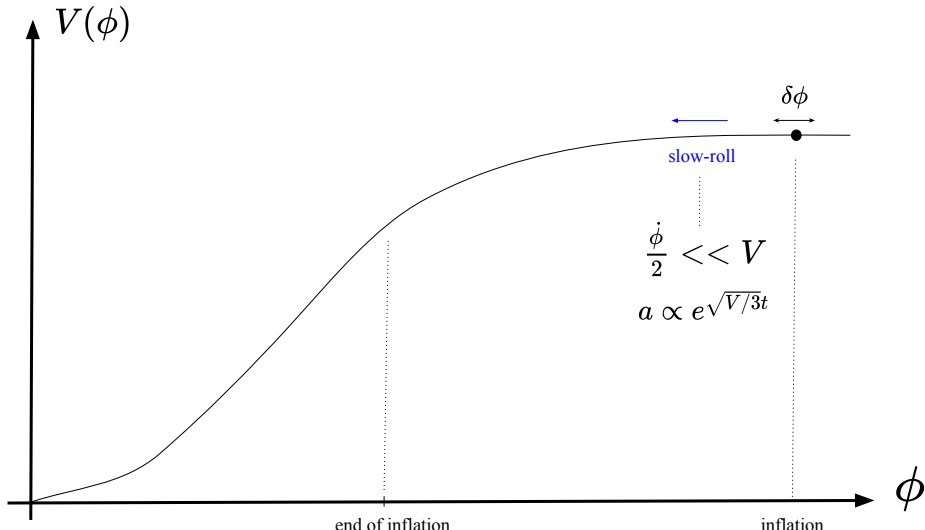

Figure 12: *Slow-roll scalar field potential $V(\phi)$ as a function of $\phi$. A Slow-roll condition implies that the potential does not vary much with the evolution of the field, allowing for accelerating solutions $w < -1/3$. In the evolution of the scalar field, time runs from right to left. At the beginning of the evolution, $V(\phi) >> \dot{\phi}^2$, implies a constant $H = \sqrt{V/3}$ and a de-Sitter solution for the scale factor. In addition, quantum fluctuations $\delta\phi$ source density scalar perturbations. Inflation ends when the slow-roll condition is not satisfied.*

For the scalar field $\phi$, the equation of motion is

$$\frac{1}{\sqrt{-g}}\partial_\mu(\sqrt{-g}\partial^\mu\phi) + V_{,\phi} = 0 \qquad \Rightarrow \qquad \ddot{\phi} + 3H\dot{\phi} + V_{,\phi} = 0, \qquad (5.14)$$

for homogeneous $\phi$ with $H^2 = \frac{1}{3}(\dot{\phi}^2/2 + V(\phi)) \approx \frac{V(\phi)}{3}$.

So far, we have discussed inflation classically. In addition, quantum fluctuations of the scalar field $\delta\phi$ and the metric $\delta g^{\mu\nu}$ source density perturbations and gravitational waves, see Secs. 5.4 and 6.2.

## 5.2 Characteristic frequencies of relic gravitational waves

If a gravitational wave is emitted with some frequency $f_*$ at time $\tau_*$, then the observed frequency at time $\tau_0$ is redshifted due to the expansion of the universe,

$$f_0 = f_* \frac{a(\tau_*)}{a(\tau_0)}. \qquad (5.15)$$

The waves are produced during *horizon re-entry* when the wavelength becomes sizable to the comoving Hubble horizon. The comoving Hubble horizon $(aH)^{-1}$ is controlled by the Hubble parameter, $H \sim f = 1/t$. The emitted frequency evolves as $f_* = (\epsilon_* H_*^{-1})^{-1}$, where $\epsilon_*$ is a small value satisfying $\epsilon_* \leq 1$ and its exact value depends on the source. The subscript $*$ denotes horizon crossing. This parametrization sets the inverse of Hubble factor $(H_*)^{-1}$ as the cosmological horizon. Gravitational waves from a source in the early universe cannot be correlated on time scales larger than $(H_*)^{-1}$; otherwise, it would break causality. The exception is cosmic inflation, which is actually motivated by such causality issues in the so-called homogeneity problem. We comment more about it in Sec. 5.4.

Assuming that a gravitational wave signal is produced during the radiation era,

$$H_*^2 = \frac{\rho_r}{3M_p^2} = \frac{\pi^2 g_* T_*^4}{90 M_p^2}, \tag{5.16}$$

where $T_*$ marks horizon re-entry at radiation domination era, in which $a \sim 1/T$ and $t \sim 1/T^2$. For relativistic degrees of freedom about $g_* \sim 100$, we have

$$f_0 \simeq 10^{-8} \epsilon_*^{-1} \left( \frac{T_*}{\text{GeV}} \right) \text{Hz}, \tag{5.17}$$

$$t_* \simeq 10^{-22} \epsilon_*^{-1} \left( \frac{1\text{Hz}}{f_0} \right)^2 \text{s}. \tag{5.18}$$

Thus, it is possible to associate the observed frequency of gravitational waves in the detectors with the epochs of the universe when such gravitational waves had been produced. By operating in different frequency ranges, gravitational wave detectors can probe separated energy scales and cosmological epochs. As shown in Table 1, in principle, we can access very high energy scales. These scales cannot be probed by other cosmological probes, for instance, those related to the CMB, to Big Bang nucleosynthesis (BBN), and large scale structures (LSS), that can probe temperatures $T_p \leq 1$ MeV.[20]

| $\epsilon_* = 1$ | $f_0$ (Hz) | $T_*$ (GeV) |
|---|---|---|
| PTA | $10^{-8}$ | 0.1 |
| LISA | $10^{-2}$ | $10^5$ |
| LVK | $10^2$ | $10^9$ |

Table 1: *Typical peak frequencies and their associated temperature of emission, expected for PTAs (pulsar time arrays) and the ground- and space-based gravitational wave interferometers (LVK and LISA, respectively) for $\epsilon_* = 1$.*

The correct correspondence between the frequency and temperature of the universe does depend on the relativistic and entropy number of degrees of freedom $g_*$ and $g_{*,S}$, as well as on the equation of the state of the universe. One such correspondence can be found in [95].

## 5.3 The gravitational wave spectrum and the cosmological redshift

Next, we relate the GW spectrum produced at a time $t$ long ago to the observable spectrum that is probed nowadays. Since the universe expands and the number of relativistic ($g_*$) and entropy ($g_{*,S}$) degrees of freedom changes across different stages of the universe, there is a cosmological redshift that must multiply the evolution of the GW spectrum. In turn, the GW energy density behaves like radiation, decaying with the fourth power of the scale factor $a(t)$.

---

[20]Take as a grain of salt that the stochastic gravitational wave signal is weak. As we see later, there is a $1/a^2$ suppression for largely redshifted sources.

Therefore,

$$
\begin{aligned}
\Omega_{\text{GW}}(t_0) &= \frac{1}{\rho_c^0} \frac{d\rho_{\text{GW}}(t_0)}{d\ln k} = \frac{\Omega_r^0}{\rho_r^0} \frac{d\rho_{\text{GW}}(t_0)}{d\ln k} \\
&= \frac{\Omega_r^0}{\rho_r^0} \frac{\rho_r(t)}{\Omega_r(t)} \frac{1}{\rho_c(t)} \left(\frac{\partial\rho_{\text{GW}}(t_0)}{\partial\rho_{\text{GW}}(t)}\right) \frac{d\rho_{\text{GW}}(t)}{d\ln k} \\
&= \frac{\Omega_r^0}{\Omega_r(t)} \left(\frac{\rho_r}{\rho_r^0}\right) \left(\frac{a(t_0)}{a(t)}\right)^{-4} \frac{1}{\rho_c(t)} \frac{d\rho_{\text{GW}}(t)}{d\ln k} \\
&= \frac{\Omega_r^0}{1} \left(\frac{g_*(t)T^4}{g_*(t_0)T_0^4}\right) \left(\frac{a(t_0)}{a(t)}\right)^{-4} \Omega_{\text{GW}}(t) \\
&= \Omega_r^0 \left(\frac{g_*(t)}{g_*(t_0)}\right) \left(\frac{a(t_0)T_0}{a(t)T}\right)^{-4} \Omega_{\text{GW}}(t) \\
&= \Omega_r^0 \left(\frac{g_*(t)}{g_*(t_0)}\right) \left(\frac{g_{*,s}(t_0)}{g_{*,s}(t)}\right)^{4/3} \Omega_{\text{GW}}(t).
\end{aligned}
\tag{5.19}
$$

In the fourth line, we assumed that the horizon re-entry occurred during the radiation domination era, in which $\Omega_r(t) = 1$, and used the expression for the energy density in Eq. 5.16. In the last line, we used conservation of entropy $S = g_{*,s}a^3T^3$. Finally, by knowing the relation between degrees of freedom, time, and frequency, we can write a final expression as

$$
\Omega_{\text{GW}}^{\text{observed}}(k) = \Omega_r^0 \left(\frac{g_*(k)}{g_*^0}\right) \left(\frac{g_{*,s}^0}{g_{*,s}(k)}\right)^{4/3} \Omega_{\text{GW}}^{\text{emitted}}(k).
\tag{5.20}
$$

## 5.4 Gravitational waves in an expanding universe

Here we explore some properties of GWs in an expanding universe. The metric of an expanding FRLW universe in cartesian coordinates is given by

$$
ds^2 = -dt^2 + a(t)^2 dx^i dx_i = -a(\tau)^2(d\tau^2 - g_{ij}dx^i dx^j),
\tag{5.21}
$$

where $\tau$ is the conformal time and $a$ the scale factor. Expanding around a flat homogeneous cosmological background $g_{ij} = \delta_{ij} + h_{ij}$, the linearized field equations are

$$
\Box \bar{h}_{ij}(\vec{x}, \tau) - \frac{2a'}{a}\bar{h}'_{ij}(\vec{x}, \tau) = -16\pi G T_{ij},
\tag{5.22}
$$

where derivatives with respect to conformal time are denoted by primes. Notice that the second term on the left-hand side vanishes for a static universe, and we recover the usual linearized Einstein gravitational wave equation (see the first lecture). By Fourier transforming and defining $\tilde{h}_\lambda \equiv a h_\lambda$, where $\lambda = +, \times$ are the two polarization modes of the gravitational wave, we can rewrite the field equations as

$$
\tilde{h}''_\lambda(\vec{k}, \tau) + \left(k^2 - \frac{a''}{a}\right)\tilde{h}_\lambda(\vec{k}, \tau) = 16\pi G a T_\lambda(\vec{k}, \tau).
\tag{5.23}
$$

We can approximate in the following two cases: $k^2 >> (aH)^2$ (sub-horizon modes) and $k^2 << (aH)^2$ (super-horizon modes), see Fig. 13.

In vacuum, for the *sub-horizon* case, 5.23 reduces to

$$
\tilde{h}''_\lambda(\vec{k}, \tau) + k^2 \tilde{h}_\lambda(\vec{k}, \tau) \approx 0,
\tag{5.24}
$$

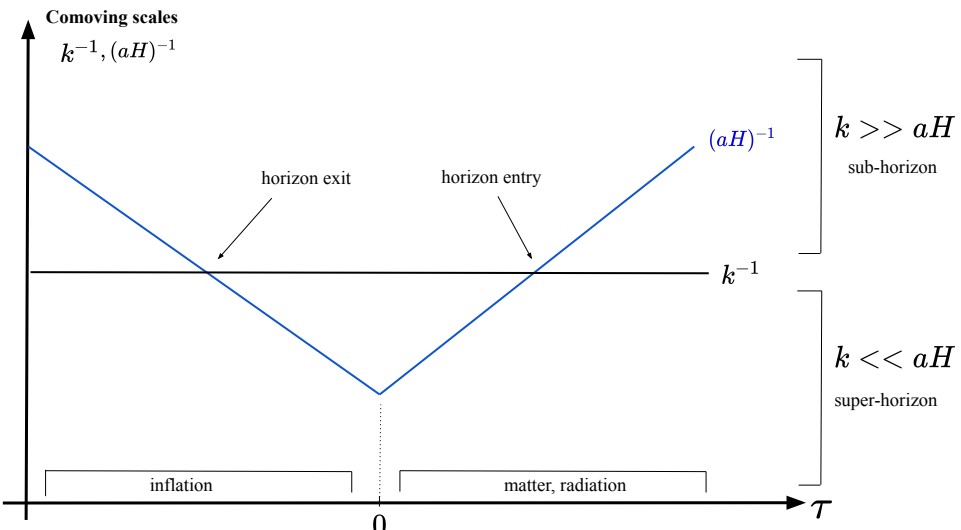

Figure 13: *Diagram of the comoving scales $k^{-1}$ and $(aH)^{-1}$ as a function of conformal time, normalized so that inflation ends at $\tau = 0$. For a mode with a given wave number $k$, at earlier times during inflation, the Hubble horizon $H$ is constant and the scale factor $a(t)$ grows exponentially. As a result, $(aH)^{-1}$ decreases, so that the mode leaves the horizon when $k^{-1} = (aH)^{-1}$. After inflation, for both matter and radiation eras, $(aH) \sim 1/t$, and $(aH)^{-1}$ increases with time so that the mode re-enters the horizon when $k^{-1} = (aH)^{-1}$. Sub-horizon scales refer to modes in the horizon $k^{-1} < (aH)^{-1}$. Super-horizon scales refer to modes out of the horizon $k^{-1} > (aH)^{-1}$. Fluctuations only propagate after horizon re-entry since they are frozen in the super-horizon.*

whose solution for $\tilde{h}$ is oscillatory. Consequently,

$$h_\lambda(\vec{k}, \tau) \approx \frac{A_\lambda}{a} \cos(k\tau + \varphi), \tag{5.25}$$

where $A_\lambda = A_\lambda(\vec{k})$ is a constant in time. Notice that wave amplitude decays as the universe expands! For the *super-horizon* case, we have instead

$$2a'h_\lambda' + ah_\lambda'' \approx 0, \tag{5.26}$$

whose solution is

$$h_\lambda = A_\lambda + B_\lambda \int_0^\tau \frac{d\gamma}{a(\gamma)^2} \approx \text{constant}, \tag{5.27}$$

where $A_\lambda = A_\lambda(\vec{k})$ and $B_\lambda = B_\lambda(\vec{k})$ are constant in time. Here we have used the fact that the integral decays in the expanding universe's history. Thus, gravitational waves are "frozen" outside the Hubble horizon. This mechanism is the same one occurring in inflation. After re-entry in the horizon, tensor perturbations become sub-horizon modes again, as described by 5.25. This is the moment when the waves are produced. Let us focus, therefore, on these sub-horizon modes.

A useful parametrization for the sub-horizon modes is [96]

$$h_{ij}^{TT}(\vec{x}, \tau) = \sum_{\lambda=+,\times} \int \frac{d^3k}{(2\pi)^3} \, h_\lambda(\vec{k}) \mathcal{T}_k(\tau) \hat{e}_{ij}^\lambda(\hat{k}) e^{-i(k\tau - \vec{k}\cdot\vec{x})}. \tag{5.28}$$

For this parametrization, we define an initial time $\tau_*$ as the time of formation or horizon entry, or for sub-horizon sources, as the time of gravitational wave emission, i.e. when the decaying behavior $(1/a)$ starts; $h_\lambda$ is the Fourier coefficient at time $\tau = \tau_*$; $\mathcal{T}_k(\tau)$ is the transfer function given here by the ratio $a(\tau_*)/a(\tau)$ (notice we have factored out $1/a$); and $\hat{e}_{ij}^\lambda$ are the components of the polarization tensor that maps the Cartesian coordinates of the tensor $h_{ij}^{TT}$ to its $+, \times$ degrees of freedom, as defined in the previous lecture.

We will use the equation for the energy density associated with gravitational waves in Sec. 3,

$$\rho_{\text{GW}}(\tau) = \frac{1}{32\pi G} \langle \dot{h}_{ij}^{TT}(\vec{x}, t) \dot{h}^{TTij}(\vec{x}, t)^* \rangle. \tag{5.29}$$

Since we are using conformal time, $\dot{h}_{ij} = (1/a)h_{ij}', \mathcal{T}_k' = -\mathcal{T}_k(a'/a)$ and $\mathcal{H} = a'/a$, for $a = a(\tau)$. Therefore,

$$\dot{h}_{ij}^{TT}(\vec{x}, t) = -\frac{1}{a} \sum_\lambda \int \frac{d^3k}{(2\pi)^3} h_\lambda(\vec{k}) \mathcal{T}_k(\tau) e_{ij}^\lambda(\hat{k})(ik + \mathcal{H}) e^{-i(k\tau - \vec{k}\cdot\vec{x})}, \tag{5.30}$$

and

$$\rho_{\text{GW}}(\tau) = \frac{1}{32\pi G} \frac{1}{a^2} \sum_{\lambda_1 \lambda_2} \int \frac{d^3k_1}{(2\pi)^3} \int \frac{d^3k_2}{(2\pi)^3} \langle h_\lambda(\vec{k}_1) h_{\lambda'}(\vec{k}_2) \rangle \hat{e}_{ij}^\lambda(\hat{k}_1) e_{ij}^\lambda(\hat{k}_2) \mathcal{T}_{k_1} \mathcal{T}_{k_2} \times$$
$$\times (ik_1 + \mathcal{H})(-ik_2 + \mathcal{H}) e^{-i(k_1 - k_2)\tau} e^{-i(\vec{k}_2 - \vec{k}_1)\cdot\vec{x}} \tag{5.31}$$

The last term simplifies after assuming homogeneity and isotropy since Eq. 4.7 holds,

$$\langle h_\lambda(\vec{k}_1) h_{\lambda'}(\vec{k}_2) \rangle = (2\pi)^3 \delta_{\lambda\lambda'} \delta^3(\vec{k}_1 - \vec{k}_2) P_\lambda(|\vec{k}_1|), \tag{5.32}$$

where $P(k)$ is the non-normalized tensor power spectrum, which has mass dimensions $[M]^{-3}$. We can explicitly factor out the powers of momentum and write the following expression with the dimensionless power spectrum of tensor perturbations introduced in Eq. 3.31, through the replacement $P_\lambda = (2\pi^2/k^3)\mathcal{P}_t(k)$.

We can write the energy density associated with SGWBs as

$$\rho_{\text{GW}}(\tau) = \frac{1}{32\pi G} \frac{1}{a^2} \sum_\lambda \int \frac{d^3k}{(2\pi)^3} \mathcal{T}_k^2 |(ik + \mathcal{H})|^2 \hat{e}_{ij}^\lambda e_{ij}^\lambda(k) P_\lambda(k). \tag{5.33}$$

For each polarization mode, $\hat{e}_{ij}^\lambda \hat{e}_{ij}^\lambda = 1$, since $\hat{e}_{ij}^{\lambda_1} \hat{e}_{ij}^{\lambda_2} = \delta_{\lambda_1, \lambda_2}$. Since we are working with *sub-horizon modes* and $|(ik + \mathcal{H})|^2 = k^2 + \mathcal{H}^2 = k^2 + (aH)^2$, we can approximate $aH << k$ so that

$$\rho_{\text{GW}}(\tau) = \frac{1}{32\pi G} \frac{1}{a^2} \sum_\lambda \int \frac{d^3k}{(2\pi)^3} \mathcal{T}_k(\tau)^2 k^2 P_\lambda(k). \tag{5.34}$$

Hence, at a time $\tau_0$:

$$\rho_{\text{GW}}(\tau_0) = \frac{1}{32\pi G} \frac{1}{a^2(\tau_0)} \frac{4\pi}{(2\pi)^3} \int (k^2 dk) \times k^2 \sum_\lambda P_\lambda(k) \frac{a^2(\tau_*)}{a^2(\tau_0)}. \tag{5.35}$$

The term $\sum_\lambda P_\lambda(k)$ can be identified with the primordial power spectrum, and $\dfrac{a^2(\tau_*)}{a^2(\tau_0)}$ tells the cosmological history. We can also relate the energy density $\rho_{\text{GW}}$ to the gravitational wave spectral shape $\Omega_{\text{GW}}$ through 3.28,

$$\rho_{\text{GW}} = \rho_c \int d(\log k) \frac{1}{\rho_c} \frac{\partial \rho_{\text{GW}}}{\partial \log k} = \rho_c \int d(\log k) \Omega_{\text{GW}}(k, \tau_0), \tag{5.36}$$

so that we can obtain the gravitational wave spectrum $\Omega_{\text{GW}}(k, \tau_0)$ by comparing the last expression with 5.35. The critical energy density is given by $\rho_c = (3H_0^2)(8\pi G)$. Therefore,

$$\Omega_{\text{GW}}(k, \tau_0) = \frac{k^2}{12H_0^2}\left(\frac{k^3}{2\pi^2}\sum_\lambda P_\lambda(k)\right)\frac{a^2(\tau_*)}{a^4(\tau_0)}. \tag{5.37}$$

Why is $\Omega_{k,\tau_0}$ relevant? Because it is the spectrum of gravitational wave density, carrying information on the source and cosmic history as measured today.

By defining the power spectra of tensor fluctuations as

$$\Delta_t^2 = \frac{k^3}{2\pi^2}\sum_\lambda P_\lambda(k) = \sum_t \mathcal{P}_t(k) \tag{5.38}$$

we can rewrite

$$\Omega_{\text{GW}}^0(k) = \frac{\Delta_t^2}{12}\frac{k^2}{H_0^2}\frac{a^2(\tau_*)}{a^4(\tau_0)} = \frac{\Delta_t^2}{12}\left(\frac{k}{a_*H_*}\right)^2\frac{a_*^4 H_*^2}{a_0^4 H_0^2}, \tag{5.39}$$

where $a_* = a(\tau_*), H_* = H(\tau_*)$ and the index 0 denotes quantities at today's time writing.

Take, for instance, the case of *single field slow-roll inflation*,[21] for $\tau_*$ in radiation domination. The power spectrum for each polarization mode is

$$P_\lambda(k) = \left(\frac{2}{M_p}\right)^2\left(\frac{H_{\text{inflation}}^2}{2k^3}\right), \tag{5.40}$$

where $M_p = 1/(8\pi G)$. We will deduce this expression later, see Eq. 6.8. It follows that $\Delta_t^2$ is constant, given by[22]

$$\Delta_t^2 = 2H_{\text{inflation}}^2/(\pi^2 M_p^2). \tag{5.41}$$

At the time of horizon entry, $a_*H_* = k$. So, $\Omega_{\text{GW}}^0$ does not depend on $k$, and the gravitational wave spectrum and the energy density decrease with $a^4$, as expected for radiation.

We can also plot the expected shape of such gravitational wave spectrum as a function of the observed frequency, see Fig. 14. If a gravitational wave is emitted with some frequency $f_*$ at time $\tau_*$ and assuming horizon sized source, then the observed frequency at time $\tau_0$ is red-shifted, see Sec. 5.2,

$$f_0 = f_*\frac{a(\tau_*)}{a(\tau_0)} \sim H_* a_*. \tag{5.42}$$

Therefore $\Omega_{\text{GW}}^0 \sim f_0^2 a_*^2$. As $H_*^2 \sim \rho$ and during radiation domination $\rho \sim a^{-4}$, then $H_* \sim a^{-2}$, $f_0 \sim a_*^{-1}$, and finally $\Omega_{\text{GW}}^0 \sim (f_0)^0$, i.e., the gravitational wave spectrum does not depend on the observed frequency. Instead, if the gravitational wave was emitted during a matter-dominated era, then $H_* \sim a^{-3/2}$, $f_0 \sim a_*^{-1/2}$, and finally $\Omega_{\text{GW}}^0 \sim f_0^{-2}$. This is the behavior expected for inflationary gravitational waves. Different primordial sources will depend on $f_0$ in different ways.

---

[21]During inflation, gravitational waves are created from quantum fluctuations in quasi-de-Sitter spaces. Here we assume that the polarization modes are the same, i.e., $P_+ = P_-$, which is a valid assumption for single-field slow-roll inflation, that shows no preference for either polarization, and the SGWB is unpolarized.

[22]In the inflationary period, $a(t) = e^{\Lambda t}$, for constant $\Lambda = H_{\text{inflation}}$. Since the $k^3$ from the power spectrum cancels the corresponding term in the definition of the tensor fluctuation, $\Delta_t^2$ is constant for any value of $k$.

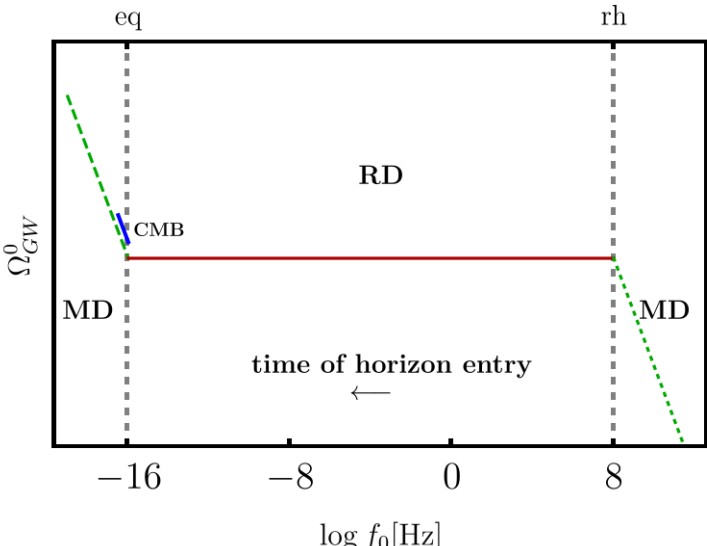

Figure 14: *Plot in logarithm scale of the gravitational wave power spectrum from single field slow-roll inflation versus frequency (observed today), in Hz. Notice that time flows in the opposite direction of the frequency axis since $f_0 \sim a_*^{-1}$. The spectrum does not depend on the frequency during the radiation-domination era (RD, red, solid line, $w_* = 1/3$). During the matter-domination era (MD, green, dashed line, $w_* = 0$), $\Omega_{\rm GW}^0 \sim f_0^{-2}$. The acronym "eq" stands for matter-radiation equality, and "rh" for reheating (the last stage of inflation, which takes into account all processes from the decay of the inflaton field in order to establish the hot thermal bath of the Big Bang). We assumed that reheating (MD, green, dotted line) occurred at $f_0 \sim 10^8 Hz$. During reheating $w_* =< w >= 0$, as MD. On left, we also show the power spectrum of CMB (blue, thick line, $r = \Delta_t^2/\Delta_s^2 < 0.1$), related to tensor anisotropies on the last scattering surface, during the matter-domination era.*

## 5.5 Constraints from BBN and CMB

Big Bang nucleosynthesis (BBN) and the cosmic microwave background (CMB) are also cosmological probes. They help us to answer the question: "What is the maximum fraction $\Omega_{\rm GW}/\Omega_r$ we can observe today?"

We start computing $\Delta\rho_r = \rho_r^{\rm obs} - \rho_r^{\rm SM}$, the extra observed radiation due to neutrino species. The energy density due to gravitational waves cannot be larger than $\Delta\rho_r$ itself. After electron decoupling,

$$\rho_r = \rho_\gamma + \rho_\nu = \frac{\pi^2}{30}\left(2 + \frac{7}{4}N_{\rm eff}\left(\frac{4}{11}\right)^{4/3}\right)T^4. \tag{5.43}$$

The factor of 2 corresponds to the two degrees of freedom of photon radiation, $7/4 = (7/8)2$ to neutrinos and anti-neutrinos (fermions with one helicity state each), and $4/11$ to heating of the photon bath relative to the neutrino bath due to $e^+e^-$ decay after neutrino decoupling. $N_{\rm eff}$ is the neutrino effective number given by $N_{\rm eff}^{\rm SM} + \Delta N_{\rm eff}$. Consequently,

$$\left(\frac{\rho_{\rm GW}}{\rho_\gamma}\right)_{T={\rm MeV}} < \frac{\rho_r^{\rm obs} - \rho_r^{\rm SM}}{\rho_\gamma} \leq \frac{7}{8}\left(\frac{4}{11}\right)^{4/3}\Delta N_{\rm eff}. \tag{5.44}$$

In the standard model, $N_{\rm eff}^{\rm SM} \simeq 3.0440$ [97, 98], which is consistent with the CMB data, $N_{\rm eff} = 2.99^{+0.34}_{-0.33}$ at the 95 % confidence level [88]. BBN [99] constrains $\Delta N_{\rm eff} \lesssim 0.5$ [100, 101].

Therefore, for $T < (T_{\text{BBN}}, T_{\text{CMB}})$, the observed ratio is bounded by Eq. 5.44,

$$\left( \frac{\rho_{\text{GW}}}{\rho_\gamma} \right)_T \lesssim 0.1 \,. \tag{5.45}$$

Today, $\Omega_\gamma^0 = \frac{\rho_\gamma^0}{\rho_c^0} \sim 10^{-5}$, and $\rho_{\text{GW}} \lesssim \Omega_\gamma \rho_c \Delta N_{\text{eff}}$ constrains the gravitational wave spectrum, $\rho_{\text{GW}} \lesssim 10^{-6} \rho_c$. Since $\rho_{\text{GW}} = \rho_c \int d(\log f) \Omega_{\text{GW}}(f)$, the BBN constraint implies, for a broad spectrum, that the observed gravitational wave spectrum today is bounded by

$$\Omega_{\text{GW}} \lesssim 10^{-6}. \tag{5.46}$$

This constraint holds for gravitational waves inside the horizon at $T_{\text{BBN}}$ and $T_{\text{CMB}}$, i.e., the gravitational waves were produced during radiation domination, before CMB decoupling. It already constrained some early universe models!

Next, we focus on different cosmological sources for stochastic gravitational waves.

# 6 Probing cosmology and BSM physics with the SGWB

In this section, we discuss some primordial sources of gravitational waves [8, 10, 36]: the cosmic gravitational microwave background, inflation, axion-inflation, scalar-induced gravitational waves, phase transitions, and cosmic strings. We do not cover in detail all of these models. Each one of these sources deserves a dedicated set of lecture notes. Instead, we aim for an executive summary, with more or less details depending on the model, by providing the main aspects of the source and summarizing the status of the field.

## 6.1 Cosmic gravitational microwave background

In the primordial plasma, photon decoupling at $T \sim$ eV leads to CMB. Likewise, gravitational waves decoupling at $T \sim M_P$ leads to cosmic gravitational microwave background (CGMB) [102, 103]. These are the earliest gravitational waves we can think of. Because of the very large value of the Planck mass $M_P \sim 10^{18}$ GeV, it is very hard to detect this CGMB:

$$\Omega_{\text{CGMB}} \sim \frac{T_{max}}{M_P} \Omega_{\text{CMB}}, \tag{6.1}$$

where $T_{max}$ is the highest temperature during radiation domination, at sub-Planckian temperatures (in general, $T_{max} \leq 10^{16}$ GeV; BBN bounds forbid $T_{max} \sim M_P$), and $\Omega_{\text{CMB}}$ is the CMB spectrum observed today. $\Omega_{\text{CGMB}}$ peaks around to 100 GHz [102], far from any current technology or planned gravitational wave experiment. See more about electromagnetic high-frequency (from MHz to THz) gravitational wave detection, for instance, in [104].

## 6.2 Gravitational waves from inflation

The field equations in vacuum have been described in Sec. 5.4,

$$\tilde{h}_\lambda''(\vec{k}, \tau) + \left( k^2 + \frac{a''}{a} \right) \tilde{h}_\lambda(\vec{k}, \tau) = 0. \tag{6.2}$$

Here, $\tilde{h}_\lambda = a h_\lambda$, for the two polarization modes. At $\tau \to -\infty$, we have the Bunch-Davies vacuum solution

$$\lim_{\tau \to -\infty} \tilde{h}_\lambda = \frac{2 e^{-ik\tau}}{\sqrt{2k}}, \tag{6.3}$$

so that

$$\tilde{h}_\lambda = \frac{2e^{-ik\tau}}{\sqrt{2k}}\left(1 - \frac{i}{k\tau}\right) \tag{6.4}$$

and

$$\langle h_\lambda(\vec{k})h_{\lambda'}(\vec{k})\rangle = (2\pi)^3 \delta_{\lambda\lambda'}\delta^3(\vec{k}-\vec{k}')\left(\frac{1}{M_p}\right)\frac{|\tilde{h}_\lambda|^2}{a^2} \tag{6.5}$$

gives us

$$\frac{|\tilde{h}_\lambda|^2}{a^2} = \frac{4}{a^2(2k)}\left(1 + \frac{1}{k^2\tau^2}\right) = \frac{4H^2}{2k^3}(1 + k^2\tau^2), \tag{6.6}$$

for the de-Sitter expanding universe solution, where $aH = -1/\tau$. The last term above can be neglected on super horizon scales, since $(aH)^{-1} < k^{-1}$. We have

$$\langle h_\lambda(\vec{k})h_{\lambda'}(\vec{k})\rangle = (2\pi)^3 \delta_{\lambda\lambda'}\delta^3(\vec{k}-\vec{k}')\left(\frac{2}{M_p}\right)^2 \frac{H_*^2}{2k^3}, \tag{6.7}$$

where $H_* = H(k\tau = 1)$ is the Hubble parameter when the mode $k$ leaves the horizon during inflation. Since gravitational waves are frozen on super-horizon scales and inflation ends before horizon re-entry, the last imprint from inflation in the gravitational wave signal comes from the time when the mode leaves the horizon. From 6.7, we get the power spectrum

$$P_\lambda(k) = \left(\frac{2}{M_P}\right)^2 \frac{H_*^2}{2k^3}. \tag{6.8}$$

The slow-roll condition implies $H \approx$ constant and a constant scale-invariant spectrum $\Delta_t^2 \propto k^3 P_\lambda(k)$. The GW spectrum is given by Eq. 5.39. How large are these tensor perturbations? CMB observations point to a tensor-to-scalar ratio $r = \Delta_t/\Delta_s < 0.1$ [88]. With $\Delta_s$ infered, the upper bound on $r$ is an upper bound on $\Delta_t$ and hence on the gravitational wave signal. This would imply $\Omega_{\text{GW}} \leq 10^{-15}$, which is too small for PTAs, LISA, or LVK to detect a signal.

Beyond the simplest scenario that we addressed here, scalar and tensor perturbations in slow-roll inflation are often modeled with a simple power-law spectrum. The smallness of the parameters makes it very difficult to detect these tensor perturbations in the CMB or indirect detection. One possible detectable scenario is to go beyond the single-power law parametrizations and demand that the power spectrum is blue-tilted at CMB and PTA scales, undergoing an inflection point in which it becomes red-tilted at intermediate scales, so that LVK and BBN bounds are respected. This scenario has been recently explored in [45, 105].

For more details on gravitational waves from inflation, see for instance the review [106].

## 6.3  Axion inflation

Although the previous slow-roll scalar field model is the dominant paradigm in inflation cosmology, the requirement of having a flat scalar potential is a challenge in particle physics. We can solve this problem by introducing a global symmetry, which protects the flatness of the potential from large radiative corrections, and it is spontaneously broken at large scales. One simple implementation is to assume that the inflaton $\phi$ is a pseudo-Nambu-Goldstone boson, whose model is invariant under *shift symmetry*: $\phi \rightarrow \phi + c$, where $c$ is a constant. The resulting pseudo-scalar field model is an axion-like particle model:

$$S = \int d^4x \left\{ \sqrt{-g}\left[\frac{R}{2} - \frac{1}{2}\partial_\mu\phi\partial^\mu\phi - V(\phi) - \frac{1}{4}F_{\mu\nu}F^{\mu\nu}\right] - \frac{\phi}{4\pi\bar{f}}F_{\mu\nu}\tilde{F}^{\mu\nu}\right\}, \tag{6.9}$$

where $F_{\alpha\beta} = \partial_\alpha A_\beta - \partial_\beta A_\alpha$, $\tilde{F}^{\mu\nu} = 1/2\epsilon^{\mu\nu\alpha\beta}F_{\alpha\beta}$ is the dual strength tensor, and the inflaton $\phi$ is coupled to the (dark) photon through a dimension five operator and the coupling $1/\bar{f}$. Shift symmetry[23] protects the flatness of $V(\phi)$, in the sense that symmetry breaking means departure of the flatness of the potential.

Next, we derive the equations of motion for $A_\mu$, $(d\tau = adt)$

$$\Box\vec{A} = -\vec{A}'' + \nabla^2\vec{A} = -\frac{\phi'}{\pi\bar{f}}\nabla \times \vec{A}, \qquad (6.10)$$

for the gauge-fixing $\nabla \cdot \vec{A} = 0$ and $A_0 = 0$. The right-hand side contains the axion-like particle term and is interpreted as a source term. With the Fourier decomposition

$$\vec{A}(\tau, \vec{x}) = \sum_{\lambda=\pm} \int \frac{d^3k}{(2\pi)^{3/2}} \left( A_\lambda(\tau, \vec{k})\vec{\varepsilon}_\lambda(\vec{k})\hat{a}_\lambda(\vec{k})e^{i\vec{k}\cdot\vec{x}} + \text{h.c.} \right), \qquad (6.11)$$

polarization vectors

$$\vec{\varepsilon}_\lambda(\vec{k}) \cdot \vec{\varepsilon}_\lambda'(\vec{k}) = \delta_{\lambda\lambda'}, \qquad \vec{\varepsilon}_\lambda(\vec{k}) \cdot \vec{k} = 0, \qquad i\vec{k} \times \varepsilon_\lambda(\vec{k}) = \lambda k \varepsilon_\lambda(\vec{k}), \qquad (6.12)$$

and creation and annihilation operators that satisfy $[\hat{a}_\lambda(\vec{k}), \hat{a}_{\lambda'}^\dagger(\vec{k})] = \delta_{\lambda\lambda'}\delta^{(3)}(\vec{k} - \vec{k}')$.

The equations of motion in de-Sitter space are

$$\left[ \frac{\partial^2}{\partial\tau^2} + k^2\left(1 \mp \frac{2\xi}{k\tau}\right) \right] A_\pm(\tau, k) = 0, \qquad \xi = \frac{\dot{\phi}}{2\pi\bar{f}H}. \qquad (6.13)$$

There is a tachyonic instability[24] for $A_+(\dot{\phi} > 0)$ if $-k\tau = k(aH)^{-1} < 2\xi$. In this case, there is an exponential gauge field production for $A_+(\dot{\phi} > 0)$ when $(aH)^{-1} \sim k^{-1}$, i.e., around horizon exit (assuming $\xi \sim \mathcal{O}(1)$). Therefore, with this model we produce gauge fields with a preferred helicity (helical gauge fields). Next, we want to understand the phenomenology behind this model.

Assuming Bunch-Davies vacuum at $\tau \to -\infty$, slow-roll inflation (constant $\xi$), the solutions are

$$A_\lambda^k(\tau) = \frac{e^{\lambda\pi\xi/2}}{\sqrt{2k}}W_{-i\lambda\xi,1/2}(2ik\tau), \qquad (6.14)$$

where we have used a Whittaker W function. These solutions are only valid under the assumption of small backreaction of the gauge field to the dynamics of the axion field. They are also known in the literature as the Amber-Sorbo solutions [107]. The gauge field backreacts to the equation of motion of the axion field via the friction term proportional to $\langle F\tilde{F}\rangle$ in

$$\ddot{\phi} + 3H\dot{\phi} + V_{,\phi} = \frac{1}{\pi\bar{f}}\langle EB\rangle, \qquad (6.15)$$

and $\xi$ itself. The largest contribution is at the end of inflation since $\xi \sim \dot{\phi}$ grows, as $\phi$ rolls down its potential, see Fig. 15. Therefore, the system can leave the regime of small backreaction, in which our analytic treatment is no longer valid and more robust techniques are needed. In general, the exponential production could overcome the exponential de-Sitter

---

[23]The last term of the action can be rewritten as $\phi F_{\mu\nu}\tilde{F}^{\mu\nu} = 2\phi\partial_\mu(\epsilon^{\mu\nu\alpha\beta}A_\nu\partial_\rho A_\sigma) = -2\partial_\mu\phi(\epsilon^{\mu\nu\alpha\beta}A_\nu\partial_\rho A_\sigma)$, which makes it explicitly shift-symmetric.

[24]In other contexts, tachyons are associated with space-like propagations (propagation speed larger than the speed of light). We highlight that in these notes, this is *not* what we mean by tachyonic instability. Instead, we mean a real exponential solution for the differential equation (6.13), which leads to the amplification of the mode $A_+$. The existence of the amplified solution does not depend on the usage of a de-Sitter space.

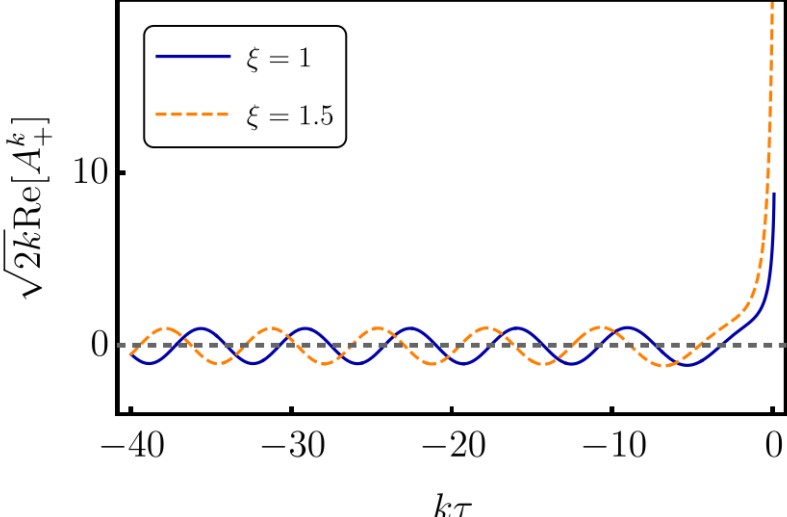

Figure 15: *Exponential gauge field production for the $A_+(\xi > 0)$ solution at around horizon exit. The larger $\xi$, the larger the gauge field production.*

expansion of the background, which dilutes matter/radiation. This way, this mechanism has phenomenological consequences, giving axion inflation signatures that can be explored by CMB observations, primordial black hole searches, and gravitational wave experiments [108]. We comment on the gravitational wave signals below.

The field equations from (6.9) for the rank-2 tensor are[25]

$$h_{ij}''(\vec{x}, \tau) + \frac{2a'}{a} h_{ij}'(\vec{x}, \tau) - \nabla^2 h_{ij}(\vec{x}, \tau) = 2\Pi_{ij}{}^{ab} T_{ab}, \tag{6.16}$$

whose solution is

$$h_{ij}(\vec{k}, \tau) = 2 \int d\tau' G_k(\tau, \tau') \Pi_{ij}{}^{ab} T_{ab}, \tag{6.17}$$

where the Green functions satisfy $[\partial_\tau^2 + 2a'/a\partial_\tau + k^2] G_k = 0$, $\Pi_{ij}{}^{ab}$ is the traceless and transverse projection operator, and $T_{ab}$ is the energy-momentum tensor from the axion-like particle model. By using the solution $A_+$ from Eq. 6.14 in $T_{ab}$, we can compute the contribution for the gravitational wave spectrum.

Since $\xi \propto \dot{\phi}$, and $\phi$ is a pseudo-scalar field, we have parity violation. The chiral contribution is a consequence of the enhancement of only the $A_+$ mode above. Indeed, we have two helicities ($h_+$ and $h_\times$) whose contributions to the gravitational wave spectrum are different. The larger contribution to the spectrum is

$$\Omega_{\text{GW}} = \left(\frac{H}{\pi M_p}\right)^2 \left(1 + 10^{-7} \frac{H^2}{M_p^2} \frac{e^{4\pi\xi}}{\xi^6}\right). \tag{6.18}$$

The first term corresponds to the vacuum contribution (solution of the homogeneous equation (6.13), without source, computed in the previous sections) and the second term is sourced by $A_\mu$. This last contribution is chiral, and its signature can be probed by LISA, the third-generation Einstein telescope, and the Cosmic Explorer detectors by searching for anisotropies. For results from the LVK collaboration, see [110], and using PTA data, see [87, 111–114].

---

[25]See details on [109].

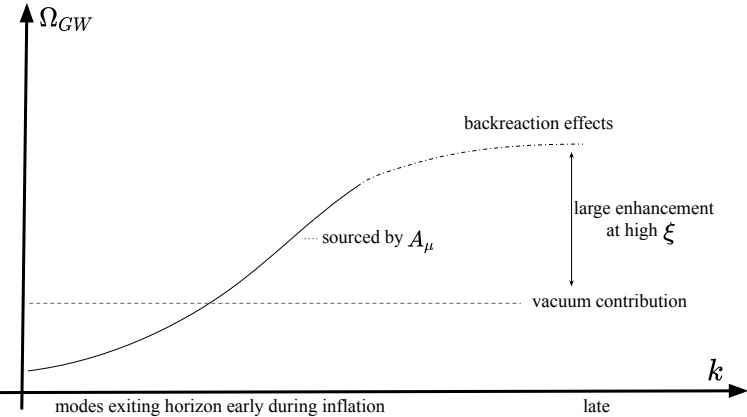

Figure 16: *The gravitational wave spectrum for the axion-inflaton model. The vacuum contribution accounts for inflationary gravitational waves without a source. Larger wavelengths exit the horizon earlier and contribute less to the spectrum since they are frozen. But there is also a large enhancement of gauge boson production, due to the velocity of the scalar field, proportional to ξ, producing helical gravitational waves. The precise balance requires taking into account the back-reaction of the gauge boson onto the axion field.*

The gravitational wave spectrum is sketched in Fig. 16. As we previously commented, there is a large enhancement for large values of $\xi$ due to the exponential production, where back-reaction effects from the gauge bosons onto the axion field become important [108]. In particular, the spectrum peaks towards higher frequencies, corresponding to modes exiting the horizon just before the end of inflation.

Since the equations of motion depart from the small backreaction regime and are highly non-linear, there are computational challenges in the field. In this regard, over the past years, we have seen continuous development in axion-inflation predictions. For instance, [108, 115, 116] showed the strongly non-linear regime experiences resonant enhancement of the gauge field production. Most recently, novel lattice techniques with a non-homogeneous axion field in [117] confirmed the previous results to some extent, despite obtaining a different behavior after the backreaction effects become important. Additionally, the gradient expansion formalism [115] is a recent and competitive alternative computation technique. Recently, a development [118] showed that the analytical regime (in which the backreaction is small) is unstable, indicating that the system may even start within the strong backreaction regime. Beyond scalar and gauge fields, non-Abelian gauge fields [119] and fermion fields could also impact axion-inflation prediction, partly suppressing the exponential production, although exact solutions are not yet known. More details on axion-inflation can be found, for instance, in [36, 120], and constraints from the latest PTA searches in [87].

## 6.4 Scalar-induced gravitational waves

Next, we describe another primordial source that is closely related to inflation: scalar-induced gravitational waves (SIGWs), which are solutions of the Einstein equations in which the tensor modes are sourced by second-order scalar modes.

In the previous sections, we discussed sources that automatically fed the tensor modes at linear order. As a result, the GW spectrum is fully characterized by the power spectrum of

tensor perturbations, as can be seen in Eq. 3.31, i.e.

$$\langle h_a(\vec{k}_1) h_b(\vec{k}_2) \rangle = \delta_{ab}(2\pi)^3 \delta^3(\vec{k}_1 + \vec{k}_2) \frac{2\pi^2}{k_1^3} \mathcal{P}_t(k_1), \tag{6.19}$$

Likewise, scalar modes are associated with the power spectrum of scalar perturbations

$$< \Phi(\vec{k}_1) \Phi(\vec{k}_2) > = (2\pi)^3 \delta^3(\vec{k}_1 + \vec{k}_2) \frac{2\pi^2}{k_1^3} \mathcal{P}_\Phi(k_1), \tag{6.20}$$

which are typically important in inflation. They cannot produce inflationary gravitational waves (these are produced by first-order tensor modes that reenter the horizon after the end of inflation) because, at linear order, scalar and tensor modes are decoupled. Rather, all modes that reentered the horizon before photon decoupling left imprints into the last scattering surface, making the CMB background a powerful probe of inflation and early-universe mechanisms that depend on these modes.

However, at higher order, scalar and tensor modes are coupled to each other. The scalar modes can source the tensor modes, inducing higher-order GWs [121,122]. From these higher-order contributions, if the scalar perturbations are large enough, the leading contribution to $\langle h_a(\vec{k}_1) h_b(\vec{k}_2) \rangle$ should be a second-order scalar-scalar contribution. This gives rise to the SIGWs.[26] The resulting GW spectrum is (see, for instance, derivation in [124,125])

$$\Omega_{\text{GW}}(k, \eta) = \frac{3}{4} \int_0^\infty dv \int_{|1-v|}^{1+v} du \left( \frac{4v^2 - (1 + v^2 - u^2)^2}{4uv} \right)^2 \frac{J(u,v)}{u^2 v^2} \mathcal{P}_\xi(vk) \mathcal{P}_\xi(uk), \tag{6.21}$$

$$J(u,v) = \left( \frac{u^2 + v^2 - 3}{2uv} \right)^4 \left[ \left( \ln \left| \frac{3 - (u+v)^2}{3 - (u-v)^2} \right| - \frac{4uv}{u^2 + v^2 - 3} \right)^2 + \pi^2 \Theta(u + v - \sqrt{3}) \right]. \tag{6.22}$$

Above, the quantity $\mathcal{P}_\xi(k)$ is the power spectrum of the comoving curvature perturbation $\xi$. The curvature perturbation is a scalar and gauge-invariant quantity, that is related to the scalar perturbation $\Phi(k)$ in Eq. 2.14 via $\Phi(k) = 2/3\xi(k)$ and is the only surviving free parameter after gauge fixing (Newton gauge) and assuming negligible anisotropic stress.

Because it is a second-order effect, the curvature perturbations must be very much enhanced to produce a detectable GW spectrum. Scenarios of ultra-slow-roll inflation could produce such enhancements, creating GWs when the perturbation modes reenter the horizon, usually assumed to occur during the radiation-domination era. Interestingly, these scalar modes can also produce *primordial black holes* (PBHs) [126,127] at horizon re-entry [128,129]. In turn, the primordial black hole population contributes to the budget of dark matter [130–132], can be seeds of the supermassive black holes at the center of galaxies [133].

Recently, we have seen an interest in the literature to fit pulsar timing data with signals from SIGW.[27] From the latest datasets, among other primordial sources, one model for SIGWs had the best performance in terms of Bayes factor [45, 87], even though the viability and

---

[26]Here, a picture with Feynman diagrams may help. We can understand the power spectra associated with the two-point functions $\langle h_a(\vec{k}_1) h_b(\vec{k}_2) \rangle$ and $\langle \Phi(\vec{k}_1) \Phi(\vec{k}_2) \rangle$ as *propagators* in quantum field theory. The first-order GW comes from $\langle h_a(\vec{k}_1) h_b(\vec{k}_2) \rangle$ computed at tree-level, in which the propagator is the inverse of the second derivative of the action with respect to the tensor modes. Second-order GWs then corresponds to the one-loop contributions to $\langle h_a(\vec{k}_1) h_b(\vec{k}_2) \rangle$. SIGW refers to the *sunset* diagram in which the two external legs are graviton lines, and the two internal lines are scalar propagators. There is no tadpole with a scalar propagator because the scalar and tensor modes are not coupled at first order, while there can also be second-order tensor-tensor (two graviton propagators) and scalar-tensor (one graviton and one scalar propagator) [123] contributions. If the scalar perturbations are sufficiently enhanced, the scalar-scalar contribution prevails.

[27]See [134] for a search with LVK data.

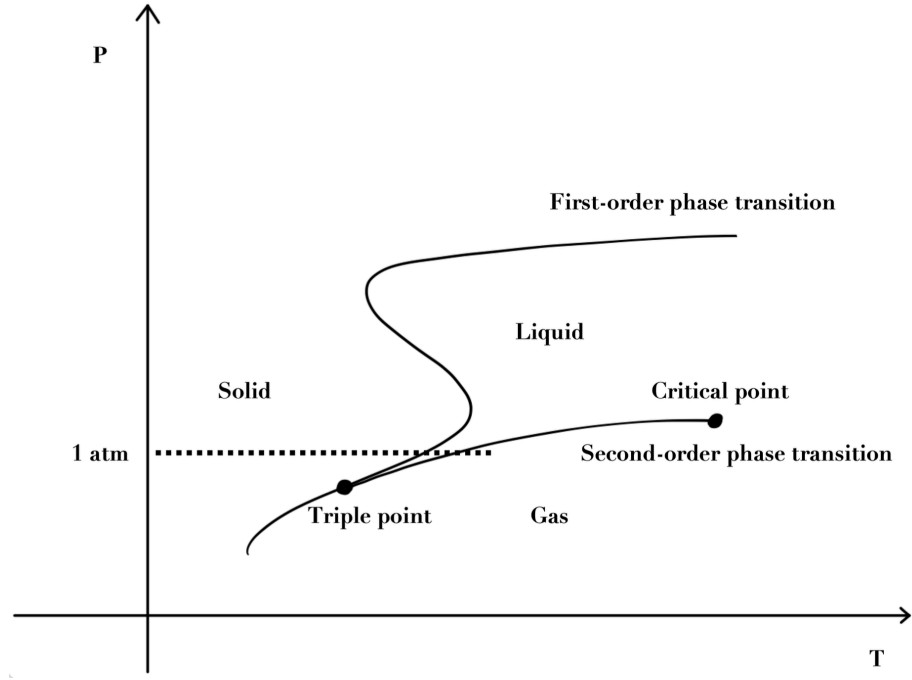

Figure 17: *Representation of the phase diagram of water as a function of pressure P and temperature T. We highlight that a second-order phase transition is associated with a critical point, in which the transition from one state to another is sudden, whereas a first-order phase transition is manifested with a continuous line.*

interpretation of these models depend on a precise formulation of the PBH DM abundance and the absence/presence of non-Gaussianities, see, for instance, [135–140]. Another interesting question is the impact of non-perturbative effects on the abundance of PBHs [141] given recent results on single-field ultra-slow-roll inflation in which the perturbativity of loop corrections are discussed (see the debate between [142] and [143], as well as their follow-up papers, that could jeopardize the SIGW interpretation).

We refer to the review [124,125] and the lecture notes [132] for more details about SIGWs.

## 6.5  First-order phase transitions

Phase transitions are observed in many physical systems. One famous example is the different physical states of a given substance – solid, liquid, or gas – according to pressure and temperature, as we can see in Fig. 17, in which we show the phase diagram for water. Depending on how the change of state occurs, we can have first-order phase transitions or second-order phase transitions. In general, they are both related to the breaking of some symmetries; the change in the thermodynamic quantities of the system (such as entropy, volume, and internal energy) is continuous for second-order phase transitions, even though their first derivatives are discontinuous, while the change is discontinuous for first-order phase transitions, associated with latent heat. In the case of water, in a liquid-gas transition at constant pressure, the temperature stays the same until all the water is boiled.

*First-order phase transitions* are processes of spontaneous symmetry breaking from a symmetric phase (false vacuum) to a broken phase (true vacuum). The symmetry breaking can naturally occur if the properties of the system change according to external parameters, such as temperature, as shown in Fig. 18. In this context, bubble collisions, magneto-hydrodynamics turbulence, and sound waves are phenomena sourcing gravitational waves during a first-order phase transition [144–154].

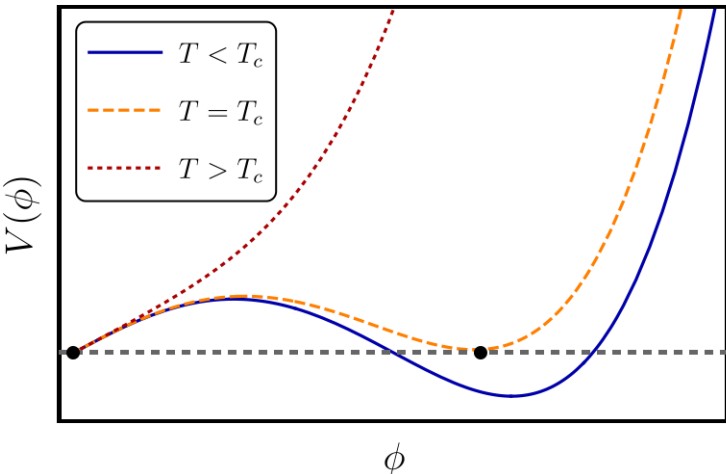

Figure 18: *Representation of a first-order phase transition, for a potential $V(\phi)$ depending on a field $\phi$ (for instance, the Higgs field) and on the temperature $T$. $\phi = 0$ is the true ground state only for $T > T_c$ (red dotted line). When the temperature decreases, a new local minimum appears, such that at $T = T_c$ both minimums are degenerate (orange dashed line). $\phi \neq 0$ is the true ground state for $T < T_c$ (blue solid line). Quantum or thermal fluctuations allow for tunneling between the vacua.*

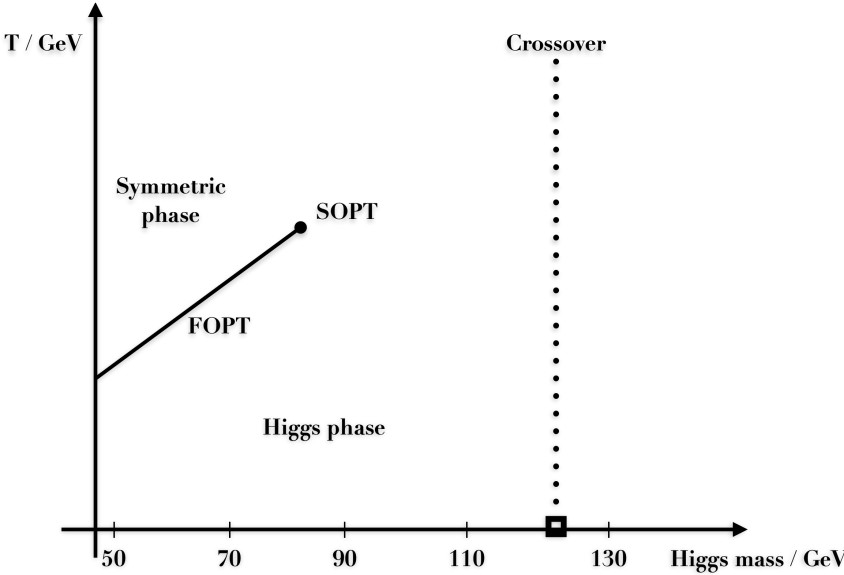

Figure 19: *Representation of the phase diagram of the SM as a function of Higgs mass and temperature. For the observed Higgs mass (small square) at around 125 GeV, we have a crossover. The Higgs is too heavy to allow the SM to undergo a first-order phase transition (FOPT) in between the symmetric and the broken (Higgs phase) phases; it is also to heavy for a second-order phase transition (SOPT).*

The peak frequency, for $\epsilon_* \sim 10^{-3}$, is around

$$f_{\text{peak}} \approx 10^{-3}\text{Hz}\left(\frac{T}{100\text{GeV}}\right), \tag{6.23}$$

where 100 GeV corresponds to the Standard Model (SM) electroweak phase transition temperature. No signal from the SM electroweak phase transition is expected because the SM does not have a first-order phase transition for the observed Higgs mass [155], instead, it has a cross-over at around 125 GeV, see Fig. 19. However, several BSM models lead to a first-order electroweak phase transition that could be probed by LISA. From BBN, $\Omega_{\text{GW}} \lesssim 10^{-6}$, already constrained some phase transition models, depending on the strength of the first-order phase transitions [156]. See also constraints from LVK in [157] and PTA searches in [45, 82, 87].

For more on cosmological phase transitions, see, for instance, the lecture notes [158].

## 6.6 Cosmic strings

Cosmic strings are one-dimensional topological defects. Topological, or cosmic, defects are products of phase transitions, when the vacuum manifold $M$ is topologically non-trivial, i.e., $\pi_n(M) \neq I$, see [159]. Here, $\pi_n(M)$ stands for the n-th homotopy group in a manifold $M$ and counts the number of equivalence classes of loops in $M$. They can be strings ($n = 1$), monopoles ($n = 2$) or textures ($n = 3$) [160]. Let us focus on strings.

Cosmic strings arise in phase transitions if, and only if, for $G \rightarrow H$, $\pi_1(G/H) \neq I$. There is no cosmic string within the SM. Cosmic string models are often associated with the spontaneous symmetry breaking of a local U(1) symmetry in some BSM/grand unified theory (GUT) scenarios [160–162]. The mechanism is sketched in Fig. 20. One example of such a BSM scenario is the breaking of $B - L$ symmetry, the difference between baryon and lepton numbers [163].

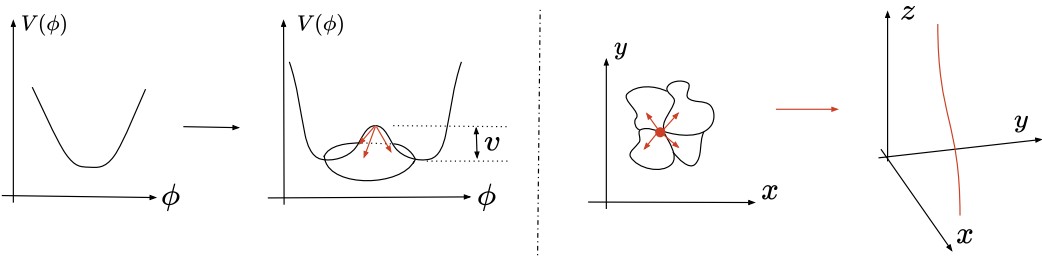

Figure 20: *Representation of cosmic strings - one-dimensional topological defects. In the two first plots, we show a complex scalar field potential versus scalar field configuration, where some mechanism allows for a phase transition with a non-vanishing expectation value v. We obtain the two last plots by mapping the solution to real space. In the 2D plot, we show the location of the local extremal point (false vacuum, orange dot) and regions where the scalar field configuration assumes different values and are causally not connected [164, 165]; by continuity, these regions intersect each other where the vacuum expectation value $< \phi >$ corresponds to the false vacuum. In the 3D plot, we extrapolate the false vacuum region to three spatial dimensions; the reason for the name* strings *becomes clear.*

In models of stable cosmic strings, the string tension $\mu$ – the energy per unit of length – is the only free parameter that characterizes the string. It is related to the amplitude of vacuum expectation value $v$ through [162]

$$v \sim \left(\frac{G\mu}{10^{-7}}\right)^{1/2} 10^{16}\text{GeV}. \tag{6.24}$$

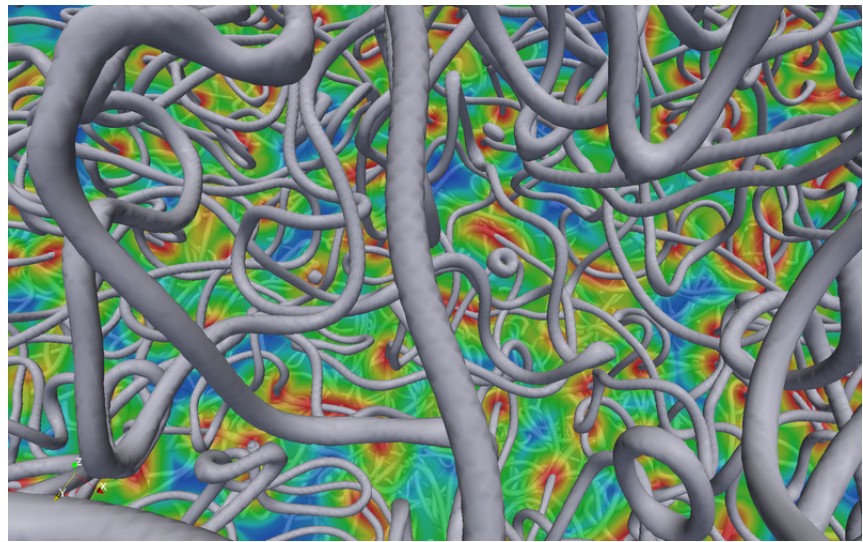

Figure 21: *3D representation of cosmic strings (gray) from a simulation credited to David Daverio, from the group of Professor Martin Kunz, Université de Genève, using simulation data obtained at the Swiss National Supercomputer Centre.*

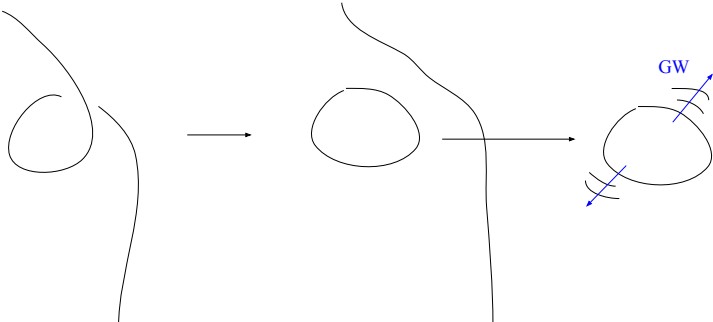

Figure 22: *Representation of gravitational waves emitted by loops from cosmic string networks.*

In the evolution of cosmic string networks, (self-)intersection generates loops. Loops are more energetically favorable, and then there is the emission of particles and gravitational waves by wave excitations of the loops. Then, most of the GW radiation is emitted by cusps and kinks [166–168]. In order to evolve the cosmic string network, heavy numerical simulation is needed and, in most cases, the strings are modeled according to the Nambu-Goto paradigm, see, for instance, [169–177]. Whether or not the Nambu-Goto strings agree with the dynamical evolution of field-theoretical cosmic strings [162,178–184] is an open question [176,177,184].

The scaling regime is a fixed point of this evolution with the property [162]

$$\frac{\rho_{\text{CS}}}{\rho_{total}} \approx \text{constant}, \tag{6.25}$$

with $\mathcal{O}(1)$ cosmic strings per Hubble volume. This property follows from the fact that in the scaling regime, the only physical scale is the Hubble radius $H^{-1}$. Therefore, the energy density of cosmic strings is $\rho = \mu \times [M]^2 \propto \mu H^2$, while the critical energy density is given by $\rho_{total} = \rho_{crit} = 3H^2/(8\pi G)$, so that the ratio $\rho_{\text{CS}}/\rho_{total}$ is constant.[28]

---

[28]This property is essential for cosmic strings phenomenology and it distinguishes strings from monopoles and domain walls. For these two-dimensional topological defects, the ratio $\rho_{\text{CS}}/\rho_{total}$ is not constant and their energy density is overproduced.

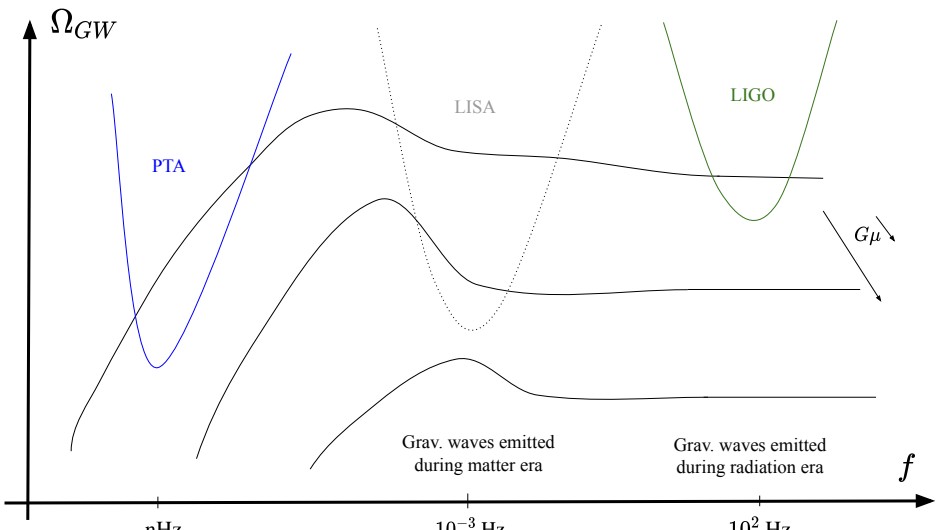

Figure 23: *Amplitude of gravitational waves generated by cosmic strings, with different string tensions μ, as a function of frequency. The larger Gμ, the larger the amplitude* $\Omega_{\mathrm{GW}}$. *We also plot the frequency range probed, or expected to be probed, by the LVK collaboration, LISA, and PTA collaborations. PTA signals already constrain cosmic string models with large Gμ, whose frequency peak is at the nHz scale.*

The gravitational wave signal is characterized by

$$\rho_{\mathrm{GW}}(t,f) \propto \sum_{n=1}^{\infty} C_n(f) P_{gw,n}. \tag{6.26}$$

In this expression, $P_{gw,n}$ is the power of a single loop. The larger the string tension $\mu$, the larger $P_{gw}$. Furthermore, $n$ corresponds to the n-th harmonic, and $C_n(f)$ gives the number of loops emitting gravitational waves, that are observed today at frequency $f$ at time $t$,

$$C_n(f) = \frac{2n}{f^2} \int dz \frac{N(l(z), t(z))}{H(z)(1+z)^6}. \tag{6.27}$$

Above, the denominator $H(z)(1+z)^6$ tells about the cosmological history. $N(l(z), t(z))$ is the number of loops of length $l$ at time $t$, where the length $l$ is given by $l = 2n/(f(1+z))$. Since it is a continuous process, the spectrum is broader. The computation is not straightforward, and there are different methods. In order to solve this integral analytically, the usual assumption is loops being sourced with $lH = \alpha = $ constant. Numerically, see, for instance, [173–175].

Fig. 23 summarizes a few properties of the GW spectrum from cosmic strings. First, the larger $G\mu$, the stronger the GW signal $\Omega_{\mathrm{GW}}$. Second, these signals are associated with a largely flat spectrum at high frequencies and with a mild peak at nHz-mHz frequencies. The flat spectrum is due to the decaying of strings produced and decaying during radiation domination while the peak corresponds to strings produced during matter domination and decaying in matter domination. Third, the larger $G\mu$, the lowest the frequency at which the spectrum peaks.

Consequently, gravitational wave searches constrain the string tension $G\mu$ and bound the symmetry-breaking scale $v$ through Eq. 6.24. For instance, large symmetry-breaking scales for topologically stable cosmic strings were already excluded by PTAs,

$$G\mu \lesssim 10^{-10} \rightarrow v \lesssim 5 \times 10^{14} \mathrm{GeV},$$

see a search for four different models of stable cosmic strings with the latest NANOGrav 15-year dataset in [45]. See also the search with EPTA dataset in [82, 185], and a recent search with NANOGrav and EPTA combined data in [87]. The string tension is also probed by the LVK collaboration [186].

Finally, as a note on possible production mechanisms, the SM cannot produce strings, but BSM theories can, such as grand unification theories (GUTs). Complementary to collider searches, gravitational waves can probe GUT physics. For instance, in some GUT models, strings can decay via monopole pair production, suppressing $\Omega_{\mathrm{GW}}$ at low $f$. *Metastable defects* relies on a sequence of phase transitions [187]

$$G \to G^{'} \to SM, \qquad \pi_n(G/SM) = \mathbb{I} \qquad \text{and} \qquad \pi_n(G/G^{'}) \neq \mathbb{I}, \pi_m(G^{'}/SM) \neq \mathbb{I}, \qquad (6.28)$$

i.e. the manifolds $(G/G^{'})$ and $(G^{'}/SM)$ have non-trivial homotopy groups, but the homotopy group of $(G/SM)$ is trivial so that the defect is not topologically stable.

For example, we have the symmetry groups $G = SO(10)$ and $G^{'} = U(1)/SM$. $G \to G^{'}$ generates monopoles, $G^{'} \to SM$ generates cosmic strings, but $\pi_1(S0(10)/SM) = \mathbb{I}$. Therefore, strings are not topologically stable, thus metastable. There are at least two decaying mechanisms. In the first mechanism, there is an initial population of monopoles and strings; then, the *string-monopole gas* decays fast. In a second mechanism, relevant if inflation dilutes away the initial monopole population, strings can only decay via spontaneous Schwinger monopole production with a decay rate $\propto e^{-m^2/\mu}$, where $m$ is the monopole mass; then, these *metastable strings* can emit gravitational waves [188]. At low frequencies, the spectrum is suppressed, and it cannot be excluded by the PTA bounds [45] while allowing for larger spectra at larger frequencies, which opens a discovery space for the LVK collaboration [189]. See a recent review on metastable strings in [190] and a search with the latest NANOGrav 15-year dataset in [45].

# 7 Conclusions

In these lecture notes on gravitational waves from the early universe, we derived the main properties of gravitational waves, introduced the stochastic gravitational wave background, discussed ongoing and future detection efforts, and introduced some primordial sources of gravitational waves.

We started with the basics in Sec. 2, with the derivation of the Einstein equations for a linear perturbed metric $g_{\mu\nu} = \eta_{\mu\nu} + h_{\mu\nu}$. We evaluated the degrees of freedom of the linearized metric tensor and fixed all the nonphysical degrees of freedom. The solution of the Einstein equation in a vacuum is a wave equation. Hence, any test mass follows a sinusoidal geodesic while a gravitational wave passes by. This effect on test masses is the gist of gravitational wave measurement.

In Sec. 3, we derived the gravitational wave emitted by a source given by an energy-momentum tensor $T_{\mu\nu}$. We showed that gravitational waves carry energy that curves the background. An important aspect of this section is the separation of scales and the relation between the gravitational wave radiation and the curved background. The power of gravitational radiation given by Einstein's quadrupole formula and the power spectrum of tensor perturbations are found in the last parts of this lecture.

In Sec. 4, we introduced the stochastic gravitational wave background (SGWB). This background is defined as the superposition of gravitational waves with different wave numbers $\vec{k}$ (both in magnitude and direction) coming from all directions in the sky. We derived their main properties. Since these waves behave like noise, it is a challenge to detect them. Boosted by the detection of gravitational waves (transient signal) by the LIGO-Virgo collaboration, a

new generation of experiments (ground-based, space-born, and pulsar time arrays) expects to detect stochastic signals in the near future. We briefly discussed experimental efforts in this direction and reported on the latest advances in PTA searches, which include evidence for Hellings-Downs correlation.

The SGWB can have astrophysical and cosmological origins. The astrophysical sources are associated with supermassive black-hole binaries. A direct detection of waves produced by these objects would confirm again a prediction of General Relativity, now for a different range of masses that ground-based interferometers cannot probe. In a binary merger, the larger the black hole masses, the lower the frequencies of the gravitational waves emitted. PTA collaborations expect to find evidence for astrophysical signals in the very near future.

On top of this astrophysical background, there are also cosmological sources from the early universe. These primordial sources produced gravitational waves way before the emission of the CMB radiation and traveled freely through the hot plasma of the early universe. Since BSM physics relies on mechanisms in energy scales beyond the ones current accelerators can probe, the phenomenology of the SGBW from the early universe is an essential step toward probing BSM physics. In this context, in Sec. 5, we introduced gravitational waves in an expanding universe and discussed how we can use data from gravitational waves as complimentary probes to the CMB and BBN bounds to constrain beyond Standard Model physics and the earlier stages of the universe.

Then, in Sec. 6, we discussed how very different early-universe sources can give rise to a background of gravitational waves. We introduce these sources chronologically according to which stage the waves are produced. This has to do with the moment that tensor perturbation re-enters the horizon. We focused on the spectrum produced by cosmic gravitational wave background (CGWB), inflationary gravitational waves, axion inflation, scalar-induced gravitational waves, first-order phase transitions, and cosmic strings.

Data from gravitational waves opened a new window to the phenomenology of new physics and can give important insights into new physics. We have already experienced how data from the latest PTA searches can really constrain new physics models. Much more is expected in the next decade, when we expect the development of detection techniques and prospects for the detection of cosmological sources from different collaborations, such as LVK, LISA, PTAs, SKA, the Einstein telescope, and the Cosmic Explorer. Better stay tuned!

Finally, we highlight that these notes at any moment intended to substitute the extensive literature on gravitational waves and the cosmological sources presented here. Instead, we believe our notes can be seen as an introduction or an executive summary of the different fields described here. This is why we highlighted aspects of some of the main cosmological sources and provided references for further reading. We encourage the readers to deepen their understanding of our main results in the original works that are available in the literature!

## Acknowledgements

We thank all the organizers and students for the friendly and productive atmosphere in the 27th W.E. Heraeus Summer School "Saalburg" for Graduate Students on "Foundations and New Methods in Theoretical Physics". We thank Valerie Domcke for her encouragement to prepare these notes, for allowing us to use her lecture notes as the starting point of this work, and for reading previous versions of the manuscript. R.R.L.d.S. thanks Richard von Eckardstein and Tobias Schröder for feedback on the manuscript, Kai Schmitz and the Particle Cosmology Münster group in the University of Münster, for discussions and hospitality during the last stages of preparation of the first version of these notes, and the NANOGrav Collaboration.

**Author contributions**    The notes were written up by Rafael R. Lino dos Santos and Linda M. van Manen. A very early version of the manuscript was based on lecture notes by Valerie Domcke, later enlarged by the authors.

**Funding information**    R.R.L.d.S. is supported by a research grant (29405) from VILLUM FONDEN. L.M.v.M is supported by the Volkswagen Foundation.

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
