# Peer review of "Gravitational waves from the early universe"

_SciPost Physics Lecture Notes_

## Round 1 · Referee Report · Anonymous (Referee 1) · 2023-3-1

Strengths

1 - The article, based on lectures given by Valerie Domcke, covers topics of great current interest in gravitational wave cosmology.
2 - It combines relevant introductory material with both theoretical and observational expositions. It is useful to have a discussion of both GW sources and their detection in one place.
3 - It conveys the excitement of the field.
4 - The grammar and presentation are generally good.

Weaknesses

1 - The pedagogical approach is rather uneven. Some topics are covered almost from an undergraduate level, and others seem to be aimed at more advanced researchers. For example, in section 2.2, the discussion of gauge invariance is begun before introducing what a gauge transform is.
2 - There are quite a few inconsistencies of notation and signs of haste in the editing. For example, if c is kept explicitly, then it should appear in the retarded time formula around 3.4.
3 - Section 5 is too brief: it doesn't address much of the physics of the production of GWs, gives little more than formulae, and in places is confusing. What does it mean, for example, to say that "loops are more energetically favourable" on p30? The process of loop production conserves energy.
4 - Section introduces the Michelson interferometer, relevant for ground-based detectors, but only mentions the detectors more relevant for cosmology (PTA and LISA).

Report

I appreciate the Saalburg summer school's encouragement of participants to write up notes from the lectures they attend. It is a useful pedagogical exercise, and I hope that it has deepened the authors' interest in the field.

However, I don't think these notes are in a good enough shape to merit publication in SciPost. The notes should pay more attention to the order in which material is introduced, its level, its clarity, and its accuracy. In order to be a useful reference complementary to those already existing, I think they should include more explanation of early universe GW production, at a level that students could then go on to start reading the research literature. It would also be useful to have more discussion of relevant detectors like LISA and PTA.

Requested changes

I don't think the notes are sufficiently close to being publishable to make a detailed list of specific changes useful.

---

## Round 2 · Referee Report · Anonymous (Referee 1) · 2023-12-23

Strengths

The authors have improved the lecture notes, and the addition of material on interferometers and pulsar timing is welcome. An explanation of the Hellings-Downs curve is very topical in view of the latest results strongly supporting the presence of a stochastic GW background.

Weaknesses

See attached file.

Report

The level of the discussion is still uneven, and there are still quite a few places, listed in the attachment, where these notes do not meet the criteria of being correct, systematic and intelligible. I regret that I cannot recommend the notes for publication in SciPost.

Attachment

---

## Round 2 · Author Response

For v2, we did a major revision in the manuscript to address the previous referee's report and to include new results from PTA searches. Sections, figures, and references have been added to increase the clarity of the work and make the discussion more uniform.

---

## Editorial Decision

resubmitted